# SyncHuman: Synchronizing 2D and 3D Generative Models for Single-view Human Reconstruction

**Wenyue Chen[1], Peng Li[2][†], Wangguandong Zheng[3], Chengfeng Zhao[2]**
**Mengfei Li[2], Yaolong Zhu[1], Zhiyang Dou[4], Ronggang Wang[1], Yuan Liu[2][†]**

[1]PKU, [2]HKUST, [3]SEU, [4]MIT

`https://xishuxishu.github.io/SyncHuman.github.io`

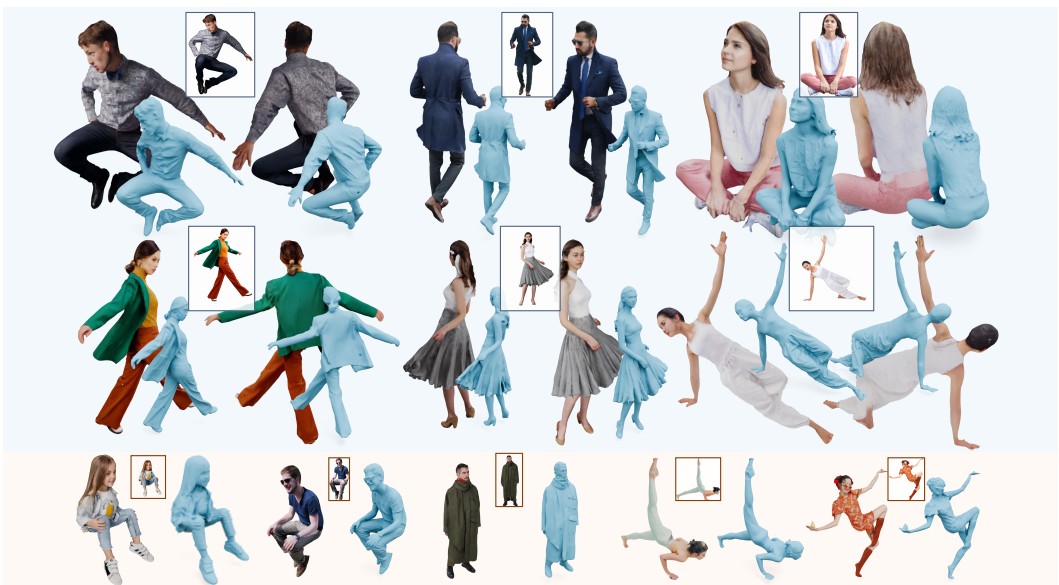

Figure 1: We introduce **SyncHuman**, a full-body human reconstruction model using synchronized 2D and 3D diffusion model. Given a single image of a clothed person, our method generates detailed geometry and lifelike 3D human appearances across diverse poses.

## Abstract

Photorealistic 3D full-body human reconstruction from a single image is a critical yet challenging task for applications in films and video games due to inherent ambiguities and severe self-occlusions. While recent approaches leverage SMPL estimation and SMPL-conditioned image generative models to hallucinate novel views, they suffer from inaccurate 3D priors estimated from SMPL meshes and have difficulty in handling difficult human poses and reconstructing fine details. In this paper, we propose SyncHuman, a novel framework that combines 2D multiview generative model and 3D native generative model for the first time, enabling high-quality clothed human mesh reconstruction from single-view images even under challenging human poses. Multiview generative model excels at capturing fine 2D details but struggles with structural consistency, whereas 3D native generative model generates coarse yet structurally consistent 3D shapes. By

---

[†]Corresponding authors

39th Conference on Neural Information Processing Systems (NeurIPS 2025).

integrating the complementary strengths of these two approaches, we develop a more effective generation framework. Specifically, we first jointly fine-tune the multiview generative model and the 3D native generative model with proposed pixel-aligned 2D-3D synchronization attention to produce geometrically aligned 3D shapes and 2D multiview images. To further improve details, we introduce a feature injection mechanism that lifts fine details from 2D multiview images onto the aligned 3D shapes, enabling accurate and high-fidelity reconstruction. Extensive experiments demonstrate that SyncHuman achieves robust and photorealistic 3D human reconstruction, even for images with challenging poses. Our method outperforms baseline methods in geometric accuracy and visual fidelity, demonstrating a promising direction for future 3D generation models.

# 1 Introduction

Reconstructing 3D clothed humans from a single RGB image is a fundamental yet challenging task. It has broad applications in AR/VR, virtual try-on, gaming, and film production [33, 36]. Compared to parametric body reconstruction [54], clothed human reconstruction [83] requires capturing not only the underlying body shape but also the diverse topology, geometry, and dynamics of garments.

Recent progress in implicit representations and generative models has led to significant advances in this area. PIFu [45] pioneered this direction with predicted neural implicit field, followed by methods such as ICON [63], ECON [62], and PaMIR [82], which introduced improvements in SMPL priors, normal estimation, and feature representation, respectively. With the advancement of generative models techniques [30, 31, 24], recent works [13, 78, 25] have introduced multiview generative model for novel-view human image prediction, enhancing 3D reconstruction fidelity, detail preservation, and robustness.

However, accurately reconstructing 3D clothed humans from a single 2D image is still challenging, especially for images with challenging poses. The reason is that most methods [13, 25] strongly rely on human shape priors, i.e., SMPL estimation, to provide structural information to generate multiview images. Unfortunately, existing single-view human pose estimation methods [7, 72, 1, 39, 3] often lack sufficient accuracy, especially when dealing with occlusions or challenging poses, as shown in Fig. 2 (a). Moreover, the estimated SMPL meshes represent only naked human bodies and fail to accurately model loose clothing. Thus, conditioned on inaccurate SMPL meshes, the multiview generative models often generate images with incorrect body topologies and mismatched details, leading to reconstruction artifacts, as shown in Fig. 2 (b).

An alternative approach employs native 3D generative models [73, 61, 27, 79] to generate the 3D human shapes directly. However, these methods often produce results lacking in detail and fidelity. By training on large-scale 3D datasets, recent 3D native generation methods [61, 79, 27] demonstrate improved capability for constructing 3D human meshes from single-view images, even for challenging human poses, as shown in Fig. 2 (c). However, these 3D native generation methods typically generate only coarse, low-fidelity shapes that poorly match the input image characteristics. Enhancing both the geometric detail and input-consistency of 3D-native generation outputs remains an open challenge.

In this paper, we propose **SyncHuman**, a novel framework that combines 2D multiview generative model and native 3D generative model for the first time, leveraging their complementary strengths to address these challenges, as shown in Fig. 2 (d). Instead of simply relying on the SMPL estimation, we utilize the more accurate

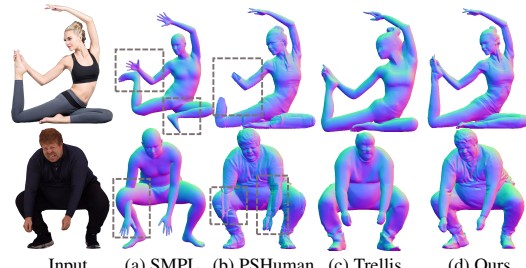

Input    (a) SMPL    (b) PSHuman    (c) Trellis    (d) Ours

Figure 2: Geometric comparison between SMPL estimation [39], 2D multiview generative model (MVD) PSHuman [25], native 3D generative model Trellis [61] and our method. 2D MVD produces high-quality details but has geometry artifacts when conditioned on inaccurate SMPL meshes. Native 3D generative model produces correct coarse structure but loses fine details and fidelity. Our method combines the strengths of both 2D and 3D generative models to produce detailed 3D human meshes with high fidelity.

3D shapes generated by the native 3D generative model to guide the generation of 2D multiview images, greatly improving the multiview consistency. At the same time, the detailed multiview images also guide the 3D generative model to carve the 3D shapes with detail and high fidelity.

In implementation, SyncHuman consists of two main components. First, we design a unified 2D-3D cross-space generative model with two branches, i.e., a 2D multiview generation branch and a 3D sparse structure generation branch, which interact via 2D-3D synchronization attention layers. The 2D-3D attention layers align the multiview images with the generated 3D shapes, which simultaneously utilize the 3D shapes to improve the cross-view consistency and employ the multiview images to enhance the fidelity of the generated 3D shapes. Next, to obtain high-quality 3D meshes, we design a multiview guided decoder to incorporate the pixel-aligned information of generated multiview images into the 3D generation during the decoding process, which not only carves fine geometric detail but also greatly improves the texture fidelity.

We conduct extensive experiments on multiple datasets to evaluate the effectiveness of SyncHuman. The results demonstrate that the proposed method outperforms previous single-view human reconstruction methods [62, 13, 25] while even achieving higher fidelity and texture quality than the large-scale 3D generative models [61] trained with datasets hundreds of times larger than ours. SyncHuman unifies 2D multiview generative model and native 3D generative model within a unified framework, enabling higher-quality image-to-3D generation with improved fidelity. This demonstrates significant potential for future 3D generation model development.

## 2 Related works

**Single image human reconstruction.** Prior to the advent of generative approaches, single-image human reconstruction primarily followed either explicit or implicit representation paradigms. Explicit methods, including voxel-based techniques [55, 83], visual hull approaches [34], and depth/normal [8, 50, 10] prediction frameworks, offer computational efficiency but often sacrifice local geometric details. The explicit normal integration in ECON [62] made a significant advancement in reconstruction robustness for the explicit paradigm. In contrast, implicit methods emerged as the dominant approach due to their continuous representation capabilities. The field was revolutionized by PIFu [45, 6, 67], which established pixel-aligned implicit functions for detailed geometry recovery from single images. Subsequent approaches enhance the robustness [11, 63, 82, 68] through parametric body model integration and additional supervision from surface normals [46] and depth information [70, 81]. Most recent works [77, 78, 18, 69, 42, 41] incorporate transformer architecture and utilize large-scale human datasets to reduce inductive bias, enhancing the generalization capability. While these methods demonstrate exceptional performance in handling complex clothing and topological variations, they remain inherently constrained by their reliance on the input image, struggling with photorealistic appearance and detail recovery.

**3D Generation.** 3D generation has been significantly advanced by generative models, which can be roughly categorized into multiview generation approaches and native 3D generation methods. Multiview generation techniques [49, 31, 30, 24, 25, 17, 56, 85, 52, 65, 64, 57, 59, 53] typically employ a two-stage pipeline: first generating consistent multiview images, followed by either optimization-based reconstruction [37] or feed-forward generation [22, 23]. The multiview generation stages involve fine-tuning an image generative model [43] or video generative models [2] by incorporating view-aware attention layers to ensure cross-view consistency. Native 3D generative models [80, 61, 27, 79, 5, 71, 60, 26] operate directly in 3D representation spaces (e.g., 3D Volume [61] or Signed Distance Field [38]), typically comprising a large variational autoencoder combined with a latent diffusion transformer (DiT) [40]. Trained on extensive 3D datasets, these models demonstrate exceptional geometric quality and strong generalization capabilities. Building upon these foundations, our work adapts and fine-tunes such a native 3D generator specifically for human body shape while preserving its generalization capacity.

**Generative Human Reconstruction** Generative models, such as Stable Diffusion, have emerged as a powerful tool for 3D human reconstruction. Pioneering works [28, 16, 15] employ score distillation sample (SDS) to optimize textured human mesh per case, which is time-consuming and typically only text-constrained. Feed-forward methods [13, 4] leverage pose-guided ControlNet [74] to predict plausible back views for neural reconstruction or Gaussian splatting, but their robustness suffers from

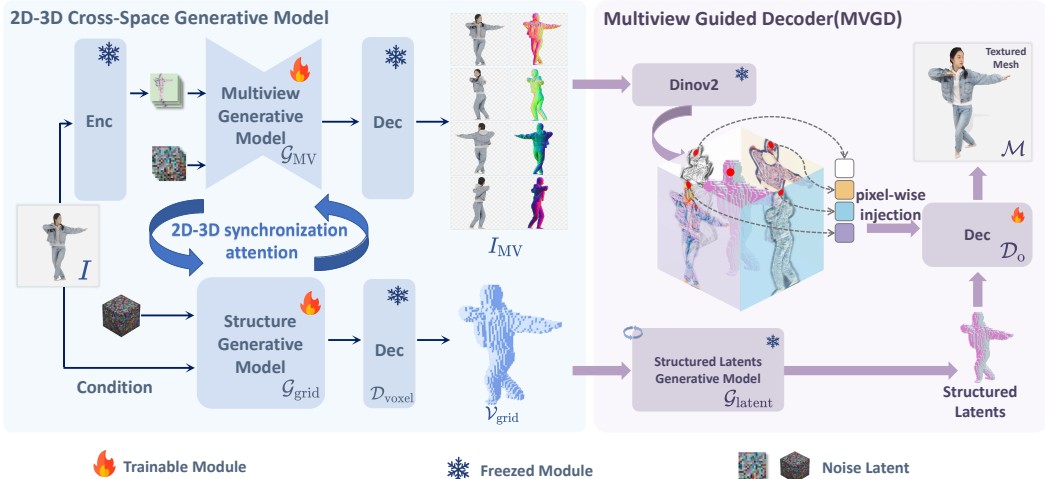

Figure 3: **Overview**. Given a single human image, SyncHuman first generates multiview color and normal maps, along with an aligned sparse voxel grid, which is further transformed into a set of structured latents. Then, we propose to inject the high-quality images into the 3D latents via a Multiview Guided Decoder and output the detailed high-fidelity textured human mesh.

limited multiview cues. Other approaches [20, 25] address this problem by fine-tuning 2D generative models to produce sparse multiview human generations. Despite improved performance, these models struggle with cross-view consistency, leading to inevitable appearance artifacts. Human3Diff [66] attempts to enhance multiview coherence by integrating 3D representations as intermediate constraints during the denoising process. However, reliance on 2D denoising generative models often leads to anatomically implausible human structures due to the absence of body prior. Unlike prior work, this study aims to align the pretrained 2D multiview and 3D native generative models, enabling producing geometrically consistent and robust 3D human models without reliance on any human prior.

## 3 Method

**Overview.** SyncHuman aims to reconstruct a 3D clothed human mesh from a single color image. As shown in Fig. 3, given a full-body human image, we first propose a 2D-3D Cross-Space generative model (Section 3.1) to synthesize multiview color and normal maps, along with an aligned sparse 3D voxel grid, which is further transformed to an aligned structured latent through a pretrained flow transformer. Then, a Multiview Guided Decoder (Section 3.2) is introduced to decode the structured latents into a high-quality, detailed, textured mesh with the help of generated multiview images.

### 3.1 2D-3D Cross-Space Generative Model

Multiview generative models have shown powerful novel-view generation and generalization capability. Given a human image as input, they could hallucinate multiple views with high-resolution details such as identity, skin texture, and clothing wrinkles, but often struggle with cross-view consistency. In contrast, native 3D generative models naturally maintain 3D structural consistency, yet typically lack fidelity. In this section, we introduce 2D-3D Cross-Space Generative Model, which combines the strengths of 2D multiview generative models and native 3D generative models.

**Multiview Generative Model.** Taking the input image $I$ as the front view, we use the network structure from PSHuman [25] to generate color and normal maps on four predefined orthogonal viewpoints, front, back, left, and right, which employs an efficient row-wise multiview attention to enhance cross-view consistency. This module could be formulated as

$$I_{\mathrm{MV}} = \mathcal{G}_{\mathrm{MV}}(I), \tag{1}$$

where $I_{\mathrm{MV}}$ is the generated multiview images and normal maps. Previous methods [25, 78, 13] usually use the estimated 3D SMPL meshes to improve the multiview consistency in $I_{\mathrm{MV}}$, but often

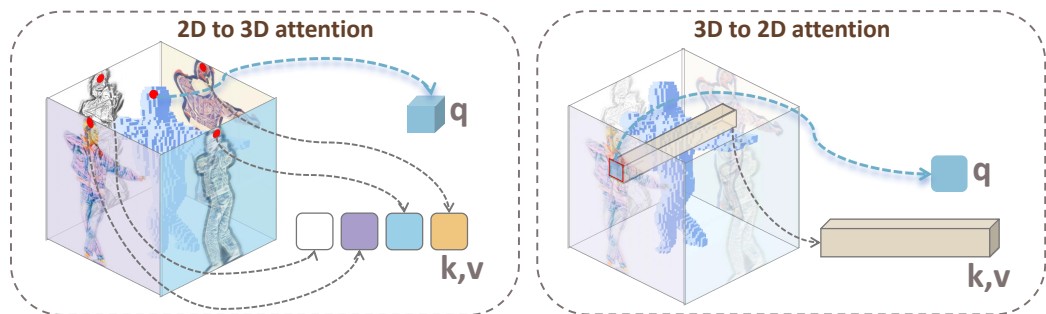

Figure 4: **2D-3D synchronization attention. 2D to 3D attention:** each 3D voxel feature is orthogonally projected onto front, back, left, and right view planes to retrieve corresponding 2D features, and refines the voxel feature with cross-attention. **3D to 2D attention:** each 2D multiview feature is projected into 3D space to attend to a column of voxel features, enhancing the 2D features. This mutual refinement ensures that 2D generative model and 3D generative model align with each other in a shared 3D space.

suffer from inaccurate SMPL estimation. Thus, we introduce the native 3D generative model to provide 3D structural guidance for the multiview generation in the following.

**3D structure Generative Model.** Our native 3D generative model follows Trellis [61]. A 3D noise grid is first used to produce a sparse structure latent through a DiT-based flow transformer $\mathcal{G}_{\text{grid}}$. The sparse structure latent is subsequently decoded into an occupied voxel grid $\mathcal{V}_{\text{grid}}$ via a Conv-based decoder $\mathcal{D}_{\text{voxel}}$,

$$\mathcal{V}_{\text{grid}} = \mathcal{D}_{\text{voxel}}\big(\mathcal{G}_{\text{grid}}(I)\big), \tag{2}$$

where the input image $I$ is fed into the generative model layer by cross-attention layers. The sparse structure generated by Trellis [61] produces reasonable 3D shapes but loses fidelity and details. We add a novel 2D-3D synchronization attention to improve the fidelity and retrieve more details from multiview images when transforming the 3D structure to textured meshes.

**2D-3D synchronization attention.** We introduce a 2D-3D synchronization attention mechanism between the 2D multiview generative model and the 3D generative model to let them benefit each other in the generation. This consists of 2D to 3D attention and 3D to 2D attention layers as follows.

**(1) 2D to 3D attention.** As shown in Fig. 4, for each 3D voxel feature, we first sample the corresponding 2D features on generated four normal maps. Then, the 3D voxel feature is used as the query token, and the concatenated 2D features from four views serve as keys and values for cross-attention. The cross-attended features are processed by an output MLP with zero initialization, and the resulting features are added to the original 3D voxel feature for refinement.

**(2) 3D to 2D attention.** Then, for each 2D feature on multiview images, we query the corresponding 3D voxel columns as in Fig. 4. Then, the 2D feature is used as the query while the 3D voxel feature serves as keys and values for cross-attention. The cross-attended features are processed by an output layer with zero initialization, and the results are added to the 2D features.

**Discussion.** Our method establishes an explicit correspondence between the 2D and 3D generative models, which benefits each branch. Through this synchronized attention, 3D generative model provides 3D structural guidance for the 2D generative model to improve the multiview consistency while the 2D generative model regularizes the 3D generative model to generate shapes that are more aligned with the input image with better fidelity. This integration enables our model to combine the advantages of both approaches: the 2D generative model provides detailed, high-fidelity results, while the 3D generative model ensures structural integrity and robust handling of complex human poses.

**2D-3D joint training.** We employ the flow matching [29] objective to train our 2D-3D cross-space generative model with the training loss defined by

$$\mathcal{L} = \left\| \boldsymbol{v}_\theta^{2d}(\boldsymbol{x}_t^{2d}, I) - (\boldsymbol{x}_0^{2d} - \boldsymbol{\epsilon}^{2d}) \right\|_2^2 + \left\| \boldsymbol{v}_\theta^{3d}(\boldsymbol{x}_t^{3d}, I) - (\boldsymbol{x}_0^{3d} - \boldsymbol{\epsilon}^{3d}) \right\|_2^2, \tag{3}$$

where $\epsilon^{2d}$ and $\epsilon^{3d}$ is the 2D noise maps and 3D noise grid, $x_t^{2d}$ and $x_t^{3d}$ is the latent features at timestep $t$ and $\boldsymbol{v}_\theta^{2d}$ and $\boldsymbol{v}_\theta^{3d}$ are the corresponding predicted velocity during denoising process, respectively. Note that the multiview generative model is based on the Stable Diffusion 2.1 [44], and we retarget it to the same flow matching model as Trellis for jointly training.

## 3.2 Multiview Guided Decoder (MVGD)

This section utilizes the generated multiview images and sparse voxels to recover textured 3D meshes.

**Structured latent generation.** We first apply another DiT-based generative model $\mathcal{G}_{\text{latent}}$ in Trellis [61], which is named as Structured Latents Generative Model in Fig. 3, to generate a set of structured latents $\mathcal{V}_{\text{latent}}$. Each of them is attached to a previously generated 3D voxel. These structured latents can be processed by either a mesh decoder $\mathcal{D}_m$ or a 3D Gaussian Splatting [19] (3DGS) decoder $\mathcal{D}_{gs}$ to generate a mesh or a 3DGS representation. For simplicity, we unify these decoders as $\mathcal{D}_o$. However, directly decoding these latent to mesh or 3DGS leads to a lack of reconstruction details, particularly noticeable in areas such as the face and clothing wrinkles, as demonstrated in Fig. 8. To address this, we propose a multiview feature injection mechanism to incorporate the generated high-resolution multiview images into the original decoder.

**Multiview feature injection.** Specifically, we extract DINOV2 [35] features of generated multiview images, and process them with several trainable MLP layers. For each 3D voxel, we query the corresponding four-view image features and concatenate them with the generated structure latent. The concatenated features are first passed through a MLP, and the resulting representations are subsequently fed into the original decoder $\mathcal{D}_o$ to produce a high-quality mesh and 3DGS representation. This simple but efficient feature injection allows for preserving the geometry fidelity and appearance realism to a great extent, as shown in Fig. 8.We render images from 3DGS and then bake onto the mesh to obtain the final textured human mesh $\mathcal{M}$. The overall decoding process can be formulated as

$$\mathcal{M} = \mathcal{D}_o\big(\mathcal{G}_{\text{latent}}(I, \mathcal{V}_{\text{grid}}), I_{\text{MV}}\big). \tag{4}$$

**Training Loss.** We train the multiview guided decoder for the 3DGS branch and the mesh branch separately. For the 3DGS branch, we use L1 loss, Structural Similarity Index (SSIM), Learned Perceptual Image Patch Similarity (LPIPS) loss between renderings and ground-truth images, and a regularization loss to avoid extremely large or small opacity. For the mesh branch, we render the foreground mask, depth maps, and normal maps from the generated 3D meshes. Then, we compute the L1 or Huber loss between the ground truth and the renderings to train the decoder. More architectural design and training details are given in the supplementary material.

## 4 Experiments

### 4.1 Experiment Setup

**Dataset.** Our models are trained on several widely used 3D human scanning datasets, including THuman2.1 [70], CustomHumans [12], THuman3.0 [51], and 2K2K [10]. To construct training images, we render 8 ground-truth images using orthographic cameras with evenly distributed azimuth angles and a fixed $0°$ elevation with a resolution of $768 \times 768$. For quantitative evaluation, we utilize 100 scans from X-Humans [47] and 150 scans from CAPE [32]. X-Humans contains 233 sequences of high-quality textured scans from 20 participants. We randomly selected 5 textured scans from each of the 20 participants in the X-Humans dataset, resulting in 100 test samples. Following ICON's partitioning criteria, we subdivide CAPE into "CAPE-FP" (50 samples) and "CAPE-NFP" (100 samples) to test the generalization ability in real-world examples. We conduct comparison with the baseline methods on the aforementioned X-Humans subset and CAPE subset, and perform ablation experiments on the same X-Humans subset.

**Metric.** To evaluate reconstruction capability, we employ three primary metrics: 1-directional point-to-surface (**P2S**), $L_1$ Chamfer Distance (**CD**), and Normal Consistency (**NC**). For geometry evaluation, we align the centers of the reconstructed mesh and the ground truth mesh and then scale them so that the coordinate range of the longest axis is 1. For appearance evaluation, we render front, back, left, and right views and compute **PSNR** [58], structural similarity index (**SSIM**) [75], and perceptual image patch similarity (**LPIPS**) [76].

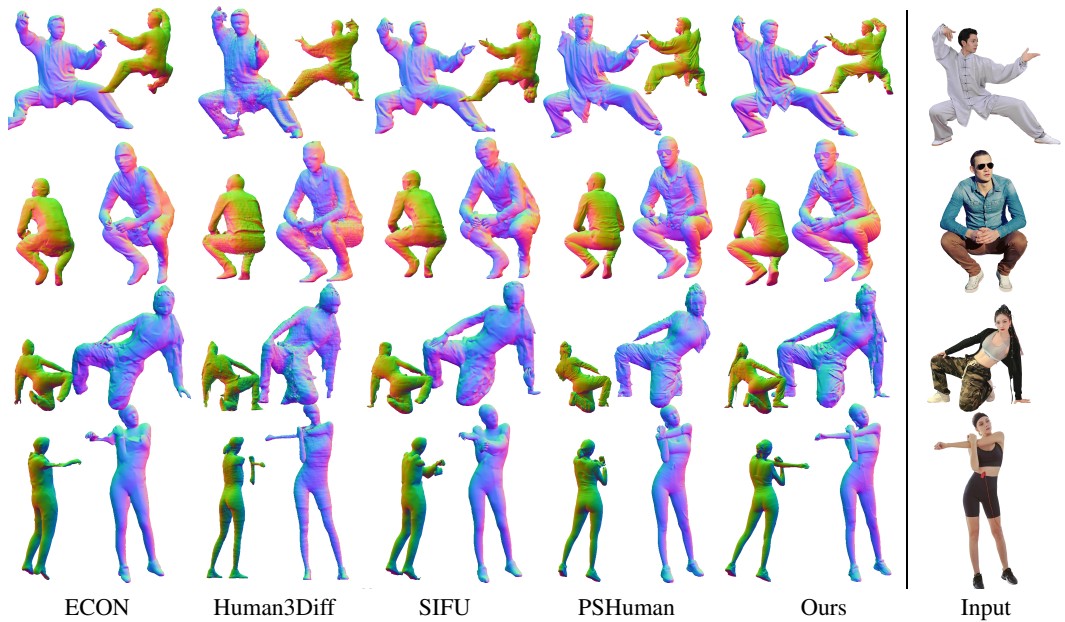

ECON      Human3Diff      SIFU      PSHuman      Ours      Input

Figure 5: Geometry comparisons between ECON [81], Human3Diff [66], SIFU [78], PSHuman [25] and ours. Our method could reconstruct 3D shapes with complete body structure and rich details.

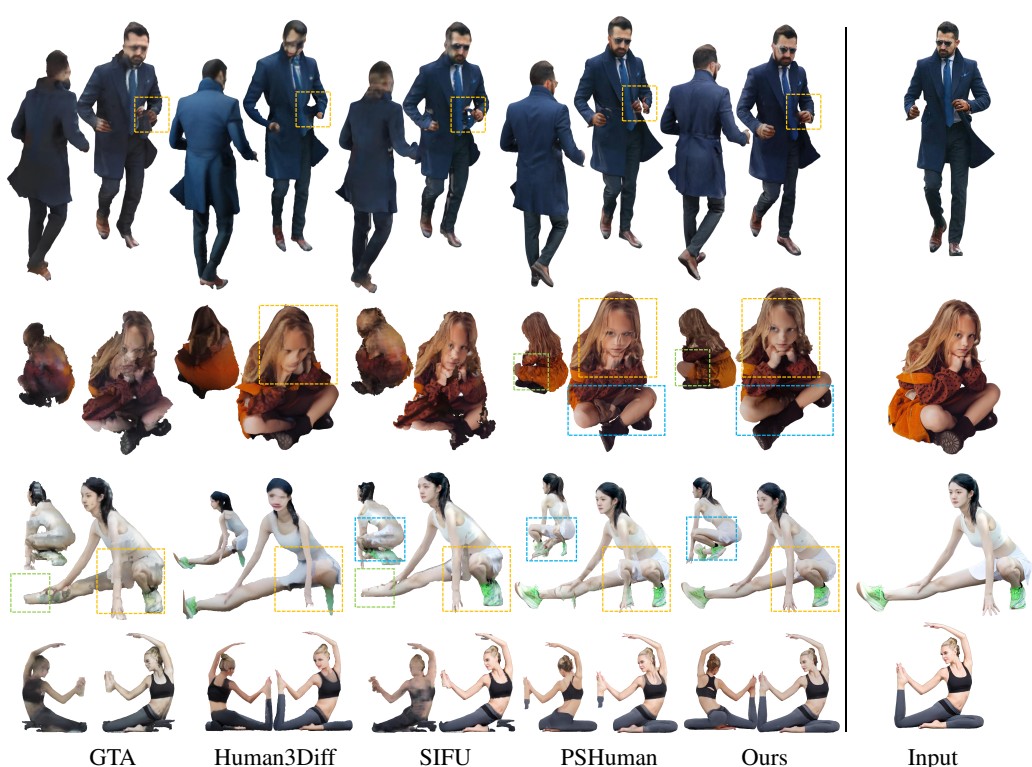

GTA      Human3Diff      SIFU      PSHuman      Ours      Input

Figure 6: Appearance qualitative comparisons between GTA [81], Human3Diff [66], SIFU [78], PSHuman [25] and our method.

Table 1: Quantitative comparison of geometry and appearance on the CAPE-NFP [32], CAPE-FP [32], X-Humans [32] datasets. Our method achieves superior performance on all metrics than baseline methods.

| Method | CAPE-NFP | | | CAPE-FP | | | X-Humans | | | | | |
|---|---|---|---|---|---|---|---|---|---|---|---|---|
| | Cham.↓ | P2S↓ | NC↑ | Cham.↓ | P2S↓ | NC↑ | Cham.↓ | P2S↓ | NC↑ | PSNR↑ | SSIM↑ | LPIPS↓ |
| ICON [63] | 1.5966 | 1.4171 | 0.7974 | 1.2698 | 1.2018 | 0.8330 | 1.4971 | 1.3920 | 0.8133 | – | – | – |
| ECON [62] | 1.8335 | 1.5391 | 0.7731 | 1.3729 | 1.2962 | 0.8225 | 1.6425 | 1.4398 | 0.8054 | – | – | – |
| GTA [77] | 1.6311 | 1.5053 | 0.7890 | 1.2980 | 1.2457 | 0.8277 | 1.5050 | 1.4662 | 0.8044 | 20.0084 | 0.8502 | 0.1129 |
| SIFU [78] | 1.6573 | 1.5130 | 0.7895 | 1.2759 | 1.2275 | 0.8289 | 1.5391 | 1.4331 | 0.8093 | 20.6747 | 0.8455 | 0.1104 |
| SiTH [13] | 1.6461 | 1.2043 | 0.7914 | 1.0377 | 0.9767 | 0.8516 | 1.5104 | 1.4345 | 0.7972 | 19.8245 | 0.8204 | 0.1182 |
| Human3Diff [66] | 1.5991 | 1.2016 | 0.7427 | 0.9666 | 0.9340 | 0.7914 | 1.5034 | 1.4219 | 0.7468 | 19.7181 | 0.8065 | 0.1334 |
| PSHuman [25] | 1.3726 | 0.9863 | 0.8276 | 0.7764 | 0.6527 | 0.8850 | 1.4377 | 1.1385 | 0.8393 | 20.8405 | 0.8523 | 0.0980 |
| TRELLIS [61] | 2.0877 | 1.5678 | 0.7521 | 1.1155 | 1.0663 | 0.8353 | 2.0043 | 1.5053 | 0.7718 | 17.0786 | 0.7238 | 0.1529 |
| OURS | 0.9127 | 0.8113 | 0.8483 | 0.6409 | 0.5962 | 0.8958 | 0.8353 | 0.7593 | 0.8872 | 21.8385 | 0.8741 | 0.0786 |

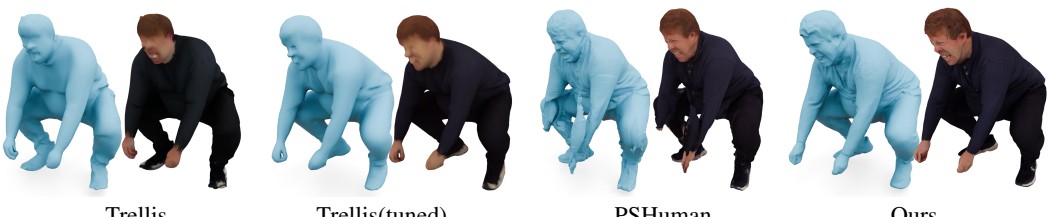

Figure 7: Ablation study of the 2D-3D synchronization attention for a joint 2D-3D modeling between PSHuman [25], fine-tuned Trellis [61], and our model.

## 4.2 Comparison with baseline methods

**Baselines.** We conducted a comprehensive comparison of our method against state-of-the-art single-view human reconstruction approaches, including classic implicit function based methods (ICON [63], GTA [77], SiFU [78]), explicit work (ECON [62]), and other baselines with generative priors ( SiTH [13], Human3Diff [66] and PSHuman [25]). All evaluations are conducted with the official open-source codes, applying a unified evaluation method. More comparisons and visual results are provided in the Appendix.

**Comparison of geometry quality.** Combining the advantages of accurate 3D coarse structure from the native 3D generative model and the rich details of the multiview generative model, our method outperforms existing approaches in geometry quality as shown in Tab. 1. The qualitative comparison in Fig. 5 highlights that our method also handles complex human poses correctly, demonstrating significant improvements in structural integrity, correctness, and detail richness over baseline methods.

**Comparison of appearance quality.** We render four views with resolution of 768 for each sample and evaluate the appearance quality by reporting average PSNR, SSIM, and LPIPS. The results presented in Tab. 1 demonstrate that our method significantly outperforms existing approaches on all metrics. As illustrated by the qualitative results in Fig. 6, our method generates high-quality appearances on novel viewpoints, delivering natural and photorealistic reconstruction quality. In contrast, existing methods exhibit notable limitations in both unseen views and occluded regions, including blurred colors and artifacts.

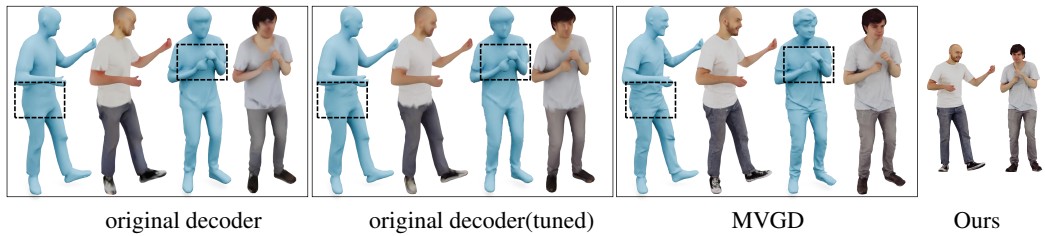

Figure 8: Ablation of different decoder settings. "original decoder" means the pretrained Trellis [61] decoder, while "original decoder (tuned)" and "multiview guided" are trained on the same human scans.All models use the same structured latents but decode them with different decoders.

Table 2: Ablation study of 2D-3D cross-space generative model on X-Humans [47] subset.

| Method | PSNR ↑ | SSIM ↑ | LPIPS ↓ | Cham. Dist ↓ | P2S ↓ | NC ↑ |
|---|---|---|---|---|---|---|
| Trellis [61] | 17.079 | 0.724 | 0.153 | 2.004 | 1.505 | 0.772 |
| Trellis [61] (tuned) | 20.344 | 0.844 | 0.101 | 1.135 | 1.041 | 0.848 |
| PSHuman [25] | 20.840 | 0.852 | 0.098 | 1.438 | 1.138 | 0.839 |
| Ours | **21.838** | **0.874** | **0.0786** | **0.835** | **0.759** | **0.887** |

Table 3: Ablation study of our multiview guided decoder (MVGD) on X-Humans [47] subset. All models employ the same structured latents but different decoders.

| Method | PSNR ↑ | SSIM ↑ | LPIPS ↓ | Cham. Dist ↓ | P2S ↓ | NC ↑ |
|---|---|---|---|---|---|---|
| original decoder | 21.083 | 0.862 | 0.092 | 0.895 | 0.820 | 0.875 |
| original decoder (tuned) | 21.362 | 0.866 | 0.090 | 0.887 | 0.810 | 0.877 |
| MVGD | **21.838** | **0.874** | **0.0786** | **0.835** | **0.759** | **0.887** |

## 4.3 Ablation Study

**2D-3D cross-space generative model.** We ablate the effectiveness of 2D-3D cross-space generative model on a X-Humans subset by removing the 2D-3D synchronization attention. For a fair comparison, we fine-tuned all the models on the same dataset. Compared with PSHuman (2D multiview generative model + remeshing) and Trellis (a native 3D generation model), this cross-space attention significantly enhances geometric accuracy and texture fidelity, as shown in Tab. 2 and Fig. 7.

**Multiview guided decoder (MVGD).** To evaluate the effect of MVGD, we compare three types of structured latent decoders: (1) the original Trellis decoder, (2) the Trellis decoder fine-tuned on human scans, and (3) the decoder guided with multiview images (our MVGD).

We conduct the comparison on the same X-Humans subset, evaluating both geometry and appearance. We apply mesh normalization and ICP registration to align the output meshes with the round truth scans to ensure a fair comparison. For each mesh, we render four views with a 768 resolution and report the average PSNR, SSIM, and LPIPS. The results in Tab. 3 show that multiview guided decoding significantly enhances geometric accuracy and texture quality. Fig. 8 also clearly illustrates that incorporating multiview image information improves the details and fidelity.

**Comparison between our structure and SMPL estimation.** To demonstrate that our structure handles some complex human poses better than the SMPL estimation, Fig. 9 shows the SMPL estimation from 4D-Humans [9] and the 3D structure generated by our method given the same input image. The SMPL estimation has obvious errors like self-intersection, while the structure generated by our method aligns better with the inputs without artifacts.

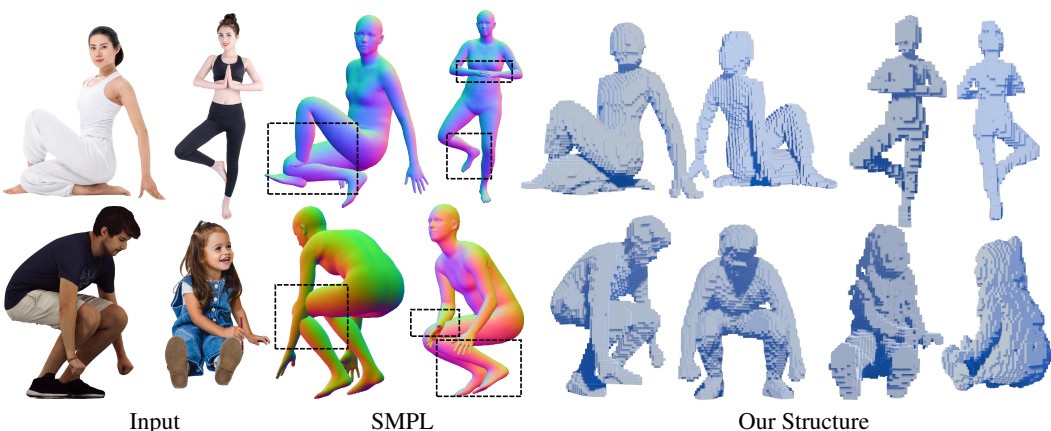

| Input | SMPL | Our Structure |
|---|---|---|

Figure 9: Robustness Analysis of the Generated Structure. The results demonstrate the robust reconstruction capabilities of our approach.

# 5 Limitation and Conclusion

In this work, we propose SyncHuman, a novel framework for robust 3D human generation from a single image. By introducing a 2D-3D cross-space generative model, we generate high-fidelity 3D structures and cross-view consistent multiview images. Then, we employ a multiview guided decoder to obtain detailed and structurally completed 3D human textured meshes. Extensive experiments

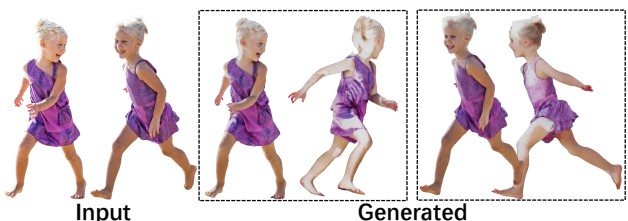

Figure 10: Unnatural textures under non-uniform lighting.

demonstrate that SyncHuman can generate 3D humans with intricate geometric details and lifelike appearances, outperforming existing methods.

**Limitations.** Our method inherits certain constraints from the training data. First, since our training dataset is rendered with uniform light source, reconstructed textures may exhibit artifacts under extreme lighting conditions (e.g., localized overexposure or shadows, as shown in Fig. 10) Moreover, our multiview generation model is fine-tuned from SD 2.1 using only ∼5,000 human scans, so its generation quality is still constrained. It will be promising to scale up our model using video generative models or large-scale multiview human datasets in future work.

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

# A Details

## A.1 Training Details

**2D-3D Cross-Space Generative Model.** Our 2D-3D Cross-Space generative model was trained on 8 NVIDIA H800 GPUs. For the multiview generative model branch, we adopt the architecture of PSHuman [25] but retrain it using flow matching from the open-source pre-trained text-to-image generation model, SD2.1-unclip [43]. We train the multiview generation branch separately with a batch size of 32 for a total of 30,000 iterations. We adopt an adaptive learning rate schedule, initializing the learning rate at 1e-4 and decreasing it to 5e-5 after 2,000 steps. For 2D-3D Cross-Space generative model, we initialize the network weights using: the fine-tuned weights from our multiview generation branch (as described above), a pre-trained image-to-3D model (Trellis [61]). Additionally, we perform zero-initialization on the output layer of the 2D-3D synchronization attention module. We train the 2D-3D Cross-Space generative model with a batch size of 32 for a total of 50,000 iterations. We adopt an adaptive learning rate schedule, initializing the learning rate at 2.5e-5 and decreasing it to 1.25e-5 after 2,000 steps. To enable class-free guidance (CFG) [14] during inference, we randomly omit the image condition at a rate of 0.05 during training.

**Multiview Guided Decoder.** Our Multiview Guided Decoder was trained on 1 NVIDIA H800 GPU. We train the decoder with a batch size of 4 for a total of 14,000 iterations, using a learning rate of 1e-4.

The loss design largely adheres to Trellis [61]'s setup. For the GS decoder, the loss includes reconstruction loss and regularization loss, with regularizations employed for the volume and opacity of the Gaussians to prevent their degeneration, specifically to avoid them becoming excessively large or transparent. $\mathcal{L}_{\text{recon}}$ is composed of $\mathcal{L}_1$ (L1 loss), Structural Similarity Index (SSIM), and Learned Perceptual Image Patch Similarity (LPIPS). The full training objective is defined as follows:

$$\mathcal{L}_{\text{GS}} = \mathcal{L}_{\text{recon}} + \mathcal{L}_{\text{vol}} + \mathcal{L}_\alpha \tag{5}$$

where:

$$\mathcal{L}_{\text{recon}} = \mathcal{L}_1 + 0.2(1 - \text{SSIM}) + 0.2 \cdot \text{LPIPS},$$

$$\mathcal{L}_{\text{vol}} = \frac{1}{LK} \sum_{i=1}^{L} \sum_{k=1}^{K} \prod s_i^k, \tag{6}$$

$$\mathcal{L}_\alpha = \frac{1}{LK} \sum_{i=1}^{L} \sum_{k=1}^{K} (1 - \alpha_i^k)^2,$$

where L is the total number of active voxels. For each active voxel, K Gaussians are predicted, $s$ and $\alpha$ are the scale and opacity of Gaussian, respectively.

For the mesh decoder, we utilize Nvdiffrast [21] to render the extracted mesh along with its attributes, producing a foreground mask $M$, a depth map $D$, a normal map $N_m$ directly derived from the mesh, an RGB image $C$, and a normal map $N$ from the predicted normals, a normal map $N_m^{\text{front}}$ directly derived from the mesh from the front view, an RGB image $C^{\text{front}}$ from the front view. The training objective is then defined as follows:

$$\mathcal{L}_{\text{M}} = \mathcal{L}_{\text{geo}} + 0.4\mathcal{L}_{\text{color}} + \mathcal{L}_{\text{reg}}, \tag{7}$$

where $\mathcal{L}_{\text{geo}}$ and $\mathcal{L}_{\text{color}}$ are written as:

$$\mathcal{L}_{\text{geo}} = \mathcal{L}_1(M) + 10\mathcal{L}_{\text{Huber}}(D) + \mathcal{L}_{\text{recon}}(N_m) + 0.1\mathcal{L}_{\text{recon}}(N_m^{\text{front}}) \tag{8}$$

$$\mathcal{L}_{\text{color}} = \mathcal{L}_{\text{recon}}(C) + \mathcal{L}_{\text{recon}}(N) + 0.1\mathcal{L}_{\text{recon}}(C^{\text{front}}) \tag{9}$$

Here, $\mathcal{L}_{\text{recon}}$ is defined identically to Eq. (7). Finally, $\mathcal{L}_{\text{reg}}$ consists of three terms:

$$\mathcal{L}_{\text{reg}} = \mathcal{L}_{\text{consist}} + \mathcal{L}_{\text{dev}} + 0.01\mathcal{L}_{\text{tsdf}}, \tag{10}$$

where $\mathcal{L}_{\text{consist}}$ penalizes the variance of attributes associated with the same voxel vertex, $\mathcal{L}_{\text{dev}}$ is a regularization.

## A.2 Detailed Network Structure

**2D-3D synchronization attention.** In the 3D branch, we inserted two 2D-to-3D attention blocks after the 8th and 16th transformer blocks respectively. Similarly, for the 2D branch, we added two 3D-to-2D attention blocks following the 3rd CrossAttnDownBlockMV2D and the UpBlock2D modules.

**(1)2D-to-3D attention.** Each 3D voxel feature $\mathbf{u}_i \in \mathbb{R}^{d_u}$ with coordinates $(x_i, y_i, z_i)$ is orthographically projected onto four view normal map planes (front, back, left, right) to obtain corresponding 2D pixel features:

$$\mathbf{p}_i^v = \pi_v(\mathbf{u}_i), \quad v \in \text{front, back, left, right} \tag{11}$$

where $\mathbf{p}_i^v \in \mathbb{R}^{d_p}$ is the projected 2D features, and $\pi_v(\cdot)$ is the orthogonal projection function.

The 3D voxel feature $\mathbf{u}_i$ and 2D pixel feature $\mathbf{p}_i^v$ are respectively passed through the MLP transformation:

$$\mathbf{q}_i = \mathrm{MLP}_q(\mathbf{u}_i) \quad \mathbf{k}_i^v = \mathrm{MLP}_k(\mathbf{p}_i^v) \quad \mathbf{v}_i^v = \mathrm{MLP}_v(\mathbf{p}_i^v) \tag{12}$$

Using 3D voxel feature as queries and concatenating four 2D pixel features along the sequence dimension as keys and values, compute the cross-attention:

$$\mathbf{K}_i = \mathrm{Concat}\left(\mathbf{k}_i^{\mathrm{front}}, \mathbf{k}_i^{\mathrm{back}}, \mathbf{k}_i^{\mathrm{left}}, \mathbf{k}_i^{\mathrm{right}}\right)$$

$$\mathbf{V}_i = \mathrm{Concat}\left(\mathbf{v}_i^{\mathrm{front}}, \mathbf{v}_i^{\mathrm{back}}, \mathbf{v}_i^{\mathrm{left}}, \mathbf{v}_i^{\mathrm{right}}\right) \tag{13}$$

$$\mathbf{u}_i' = \mathbf{u}_i + \mathrm{MLP}\left(\mathrm{Softmax}\left(\frac{\mathbf{q}_i \mathbf{K}_i^\top}{\sqrt{d}}\right)\mathbf{V}_i\right)$$

Here, $\mathbf{u}_i'$ represents the updated 3D voxel feature.

**(2)3D-to-2D attention.** Let the input consist of 2D pixel features $\mathbf{p}_i \in \mathbb{R}^{d_p}$ from a color map or a normal map, with a corresponding 3D voxel space represented by $\mathbf{U} \in \mathbb{R}^{X \times Y \times Z \times d_u}$, where $d_p$ and $d_u$ denote the feature dimensions of 2D pixels and 3D voxels, respectively.

Each 2D pixel feature $\mathbf{p}_i$ corresponds to a ray in a 3D space. Sampling $H$ 3D voxel features along this ray forms a 3D voxel feature sequence:

$$\mathcal{U}_i = \{\mathbf{u}_{i,j}\}_{j=1}^H, \quad \mathbf{u}_{i,j} \in \mathbb{R}^{d_u} \tag{14}$$

Where $\mathbf{u}_{i,j}$ is the 3D voxel feature at the $j$-th depth position along the projection ray of 2D pixel feature $\mathbf{p}_i$. $H$ is the length of the 3D voxel feature sequence.

The 2D pixel feature $\mathbf{p}_i$ is mapped to a query vector, while each 3D voxel feature $\mathbf{u}_{i,j}$ is mapped to a key and a value vector:

$$\mathbf{q}_i = \mathrm{MLP}_q(\mathbf{p}_i) \quad \mathbf{k}_{i,j} = \mathrm{MLP}_k(\mathbf{u}_{i,j}) \quad \mathbf{v}_{i,j} = \mathrm{MLP}_v(\mathbf{u}_{i,j}) \tag{15}$$

By concatenating all key vectors and value vectors across the sequence of 3D voxel features, we construct the complete key and value matrices as:

$$\mathbf{K}_i = [\mathbf{k}_{i,1}, \mathbf{k}_{i,2}, \ldots, \mathbf{k}_{i,H}] \in \mathbb{R}^{H \times d}, \quad \mathbf{V}_i = [\mathbf{v}_{i,1}, \mathbf{v}_{i,2}, \ldots, \mathbf{v}_{i,H}] \in \mathbb{R}^{H \times d} \tag{16}$$

Compute the attention output with the 2D pixel feature $\mathbf{q}_i$ as the query, and the 3D voxel feature $\mathbf{K}_i$ and $\mathbf{V}_i$ as the key and value:

$$\mathbf{p}_i' = \mathbf{p}_i + \mathrm{MLP}\left(\mathrm{Softmax}\left(\frac{\mathbf{q}_i \mathbf{K}_i^\top}{\sqrt{d}}\right)\mathbf{V}_i\right) \tag{17}$$

Here, $\mathbf{p}_i'$ represents the updated 2D pixel feature.

**Multiview Guided Decoder (MVGD).** The 3d sparse structure $\mathcal{V}_{\mathrm{grid}}$ is first processed by the Structured Latents generative model, which denoises it into a structured latent $\mathbf{z}$. For multiview color and normal images, we first upsample the images, then we extract multilevel local patch features using the DINOv2 backbone from layers $l \in \{4, 11, 17, 23\}$ per image:

$$\mathbf{F}_i^{(l)} = \mathrm{DINOv2}_l(\mathbf{I}_i^{\mathrm{up}}) \in \mathbb{R}^{V \times V \times d}. \tag{18}$$

The features from different layers are concatenated and then processed through a MLP to form the final representation:

$$\mathbf{F}_i = MLP(\mathrm{Concat}\left(\mathbf{F}_i^{(4)}, \mathbf{F}_i^{(11)}, \mathbf{F}_i^{(17)}, \mathbf{F}_i^{(23)}\right)) \in \mathbb{R}^{V \times V \times d} \tag{19}$$

Given a voxel position $\mathbf{p} = (x, y, z)$, its projection onto the $i$-th view yields the corresponding pixel coordinates:

$$\pi_i(\mathbf{p}) = (u_i, v_i), \quad \text{where } u_i, v_i \in 0, 1, \ldots, V - 1. \tag{20}$$

The corresponding image features are then retrieved via direct indexing:

$$\mathbf{f}_i(\mathbf{p}) = \mathbf{F}_i[u_i, v_i] \in \mathbb{R}^d. \tag{21}$$

The injection feature is constructed by concatenating the structured latent at position $\mathbf{p}$ ($\mathbf{z}_{\mathbf{P}} \in \mathbb{R}^{d_z}$) with multiview pixel-aligned features obtained from the color and normal maps of 4 views (8 feature vectors in total). Formally,

$$\mathbf{z}_{\mathrm{inj}} = \mathrm{Concat}(\mathbf{z}_{\mathbf{P}}, \mathbf{f}_1(\mathbf{p}), \ldots, \mathbf{f}_8(\mathbf{p})) \in \mathbb{R}^{d_z + 8 \times d}. \tag{22}$$

This injection feature is then processed by an MLP to refine the structured latent representation:

$$\mathbf{z}_{\mathbf{P}}' = \mathbf{z}_{\mathbf{P}} + \mathrm{MLP}(\mathbf{z}_{\mathrm{inj}}) \in \mathbb{R}^{d_z}. \tag{23}$$

We insert a multiview injection module after each self-attention in the decoder. We apply the same multiview feature injection mechanism to both the mesh decoder and the GS decoder, resulting in refined mesh and GS representations.

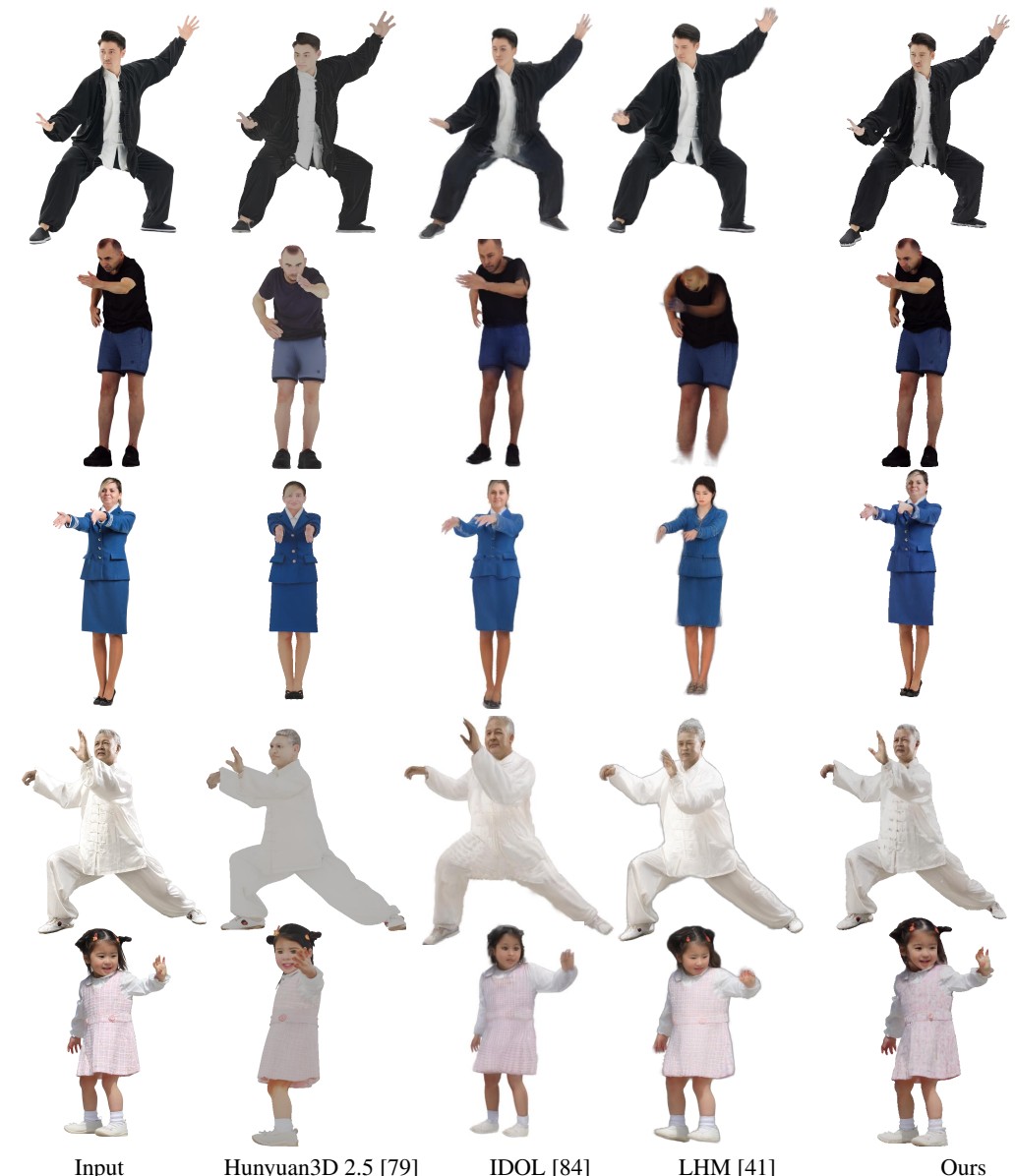

| Input | Hunyuan3D 2.5 [79] | IDOL [84] | LHM [41] | Ours |

Figure 11: Qualitative comparison of SyncHuman with Gaussians-based methods (LHM, IDOL) and a native 3D model (Hunyuan3D 2.5). SyncHuman achieves visually high-fidelity results.

# B  More Experiment

## B.1  More Results

**Comparison with Gaussians-based Methods and Native 3D generative model.** To further evaluate the effectiveness of our method, we conduct a qualitative comparison between SyncHuman and two Gaussians-based methods (LHM [41] and IDOL [84]), as well as a more advanced native 3D model, Hunyuan3D 2.5 [79], as shown in Fig. 11. All these methods are capable of producing structurally plausible and visually reasonable results. Since LHM and IDOL are based on Gaussians, they can only produce RGB images through rendering. For comparison, we render RGB images from the front view. Both LHM and IDOL rely on SMPL, and when SMPL estimation is inaccurate or fails, the resulting structure is correspondingly erroneous. Furthermore, as illustrated in Fig. 11, IDOL and LHM still exhibit limited fidelity. Hunyuan3D 2.5, trained on a large-scale dataset, is a native 3D model that can also produce reasonable human structures with details. However, as observed in Fig. 11, Hunyuan3D 2.5 produces human meshes with less fidelity.

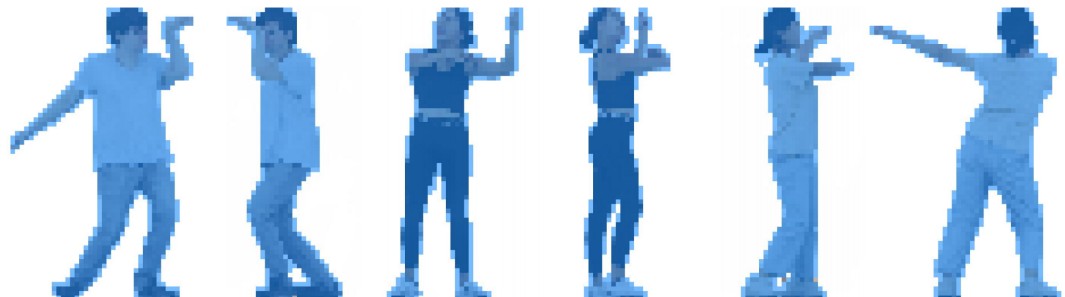

Figure 12: After alignment using 2D-3D attention, the multi-view projections of the two branches can almost completely overlap.

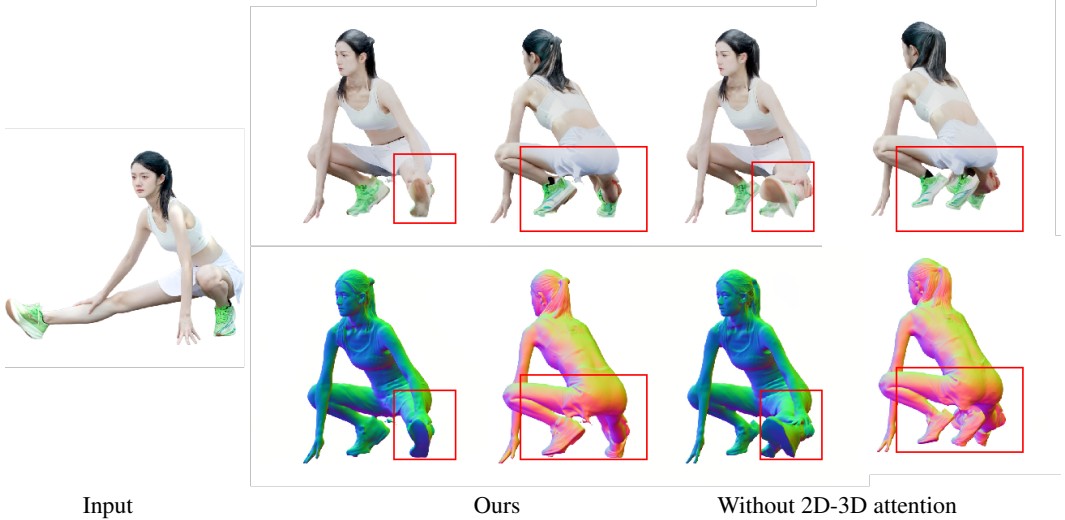

Input          Ours          Without 2D-3D attention

Figure 13: Comparison of the quality of intermediate multi-view generation with and without 2D-3D attention.

## B.2 Ablation of the quality of intermediate multi-view generation and 3D structure generation

We additionally report the quality of intermediate multi-view generation and 3D structure generation on a small human scan subset. IOU of the 3D structure generation: 0.5907 (with 2D-3D synchronization attention) vs. 0.4813(without 2D-3D synchronization attention). Color and normal image quality improvements by 3D-2D attention:

Table 4: Comparison of different methods.

| Method | PSNR↑ | SSIM↑ | LPIPS↓ |
|---|---|---|---|
| w/o att (color) | 23.328 | 0.877 | 0.078 |
| ours (color) | 24.027 | 0.894 | 0.070 |
| w/o att (normal) | 22.851 | 0.866 | 0.097 |
| ours (normal) | 23.439 | 0.882 | 0.087 |

Because generating 3D structures or multiview images from single-view inputs has ambiguity, the generation results are not exactly the same as the ground-truth. However, our 2D-3D attention could produce results more similar to GT. This demonstrates that our 2D-3D synchronization attention could benefit both branches to improve the multiview generation quality and 3D structure quality. As in Fig. 12, after alignment using 2D-3D attention, the multi-view projections of the two branches can almost completely overlap. And as in Fig. 13, when with 2D-3D attention multiview images have a more reasonable human body structure.

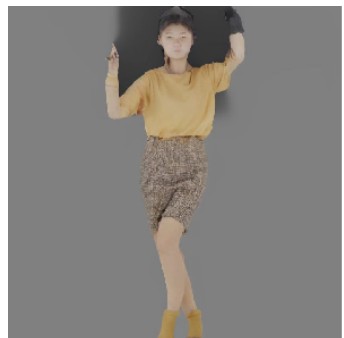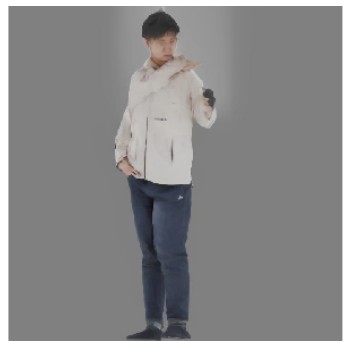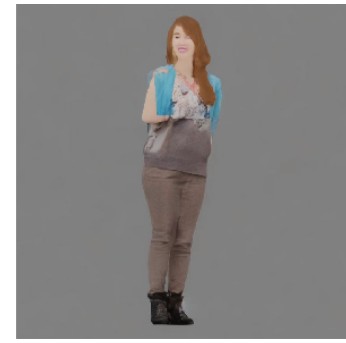

Figure 14: Visualization of the unconditional generation task.

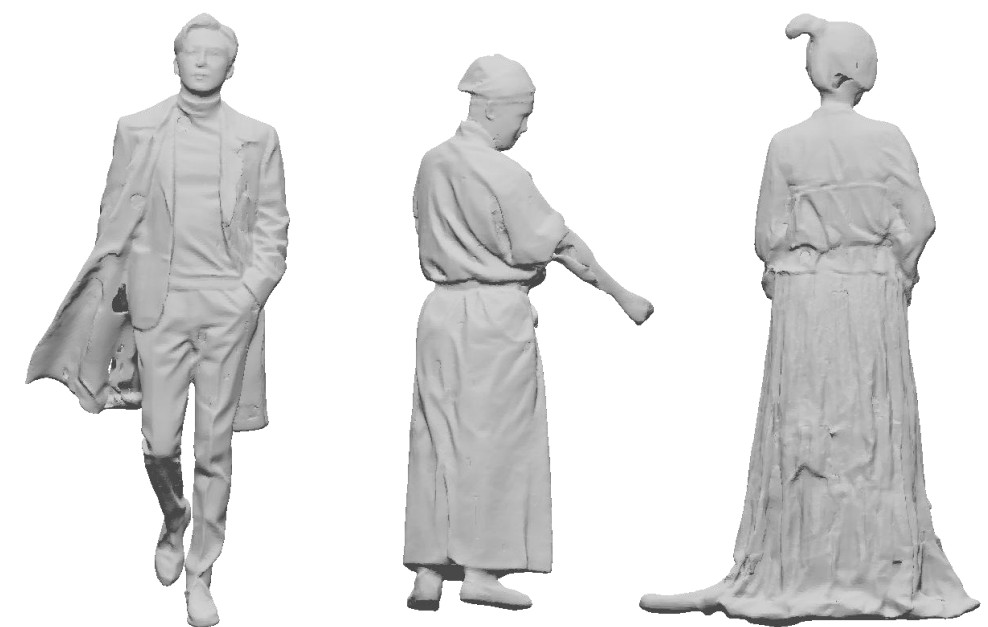

Figure 15: In some cases, the decoded mesh may contain some holes on the surface.

### B.3 Inference time.

On a single H800, the inference time is as follows: ours 38.57s vs Trellis 15.68s vs PSHuman 52.98s Our method is faster than PSHuman as it directly decodes a 3D shape without requiring additional differentiable rendering optimization. The slower speed compared to Trellis is due to our use of the 2D multi-view generation.

### B.4 Unconditional Generation.

We tested our model on the unconditional generation task as in Fig. 14. The generation quality is worse than the conditional generation from a single-view image.

## C  discussion

### C.1 Limitations about containing holes in the generation

Our method is based on Trellis [61], which uses FlexiCube [48] in the trellis mesh decoder branch does not put a water-tight constraint on the surfaces. Thus, holes may appear on the surface in some cases, as shown in Fig. 15. A possible way to make the generated meshes water-tight is to adopt another SDF fitting on the generated mesh. Alternatively, we may adopt other 3D native generative models using SDFs as targets, like Hunyuan3D [79] or TripoSG [27], to avoid this problem. We leave this for future work.

## C.2  Differences from SyncDreamer.

Our method fundamentally differs from SyncDreamer in the following two aspects. First, the synchronization subjects are totally different. SyncDreamer synchronizes the generation of multiview 2D images, whereas our method synchronizes the 2D generative model and the 3D native generative model. Our method demonstrates that simultaneously generating multiview images and 3D representations benefits each other and greatly improves the 3D generation quality. We inject information in both directions, 2D to 3D and 3D to 2D. Second, the functionality and design of the volume in our method are fundamentally different from those in SyncDreamer. SyncDreamer constructs a feature volume to share information among different views. In contrast, in our method, the volume is a meaningful 3D representation generated from noise.

## C.3  Ethics Statement

The objective of SyncHuman is to equip users with a powerful tool for creating realistic clothed 3D human models. By enabling 3D human generation from a single image, our method supports diverse ethnicities and populations, promoting equitable cultural representation. Our model was trained on the public datasets THuman2.1 [70], CustomHumans [12], THuman3.0 [51], and 2K2K [10], and tested on X-Humans [47] and CAPE [32]. However, there is a potential risk that these generated models could be misused to deceive viewers (e.g., adult content, political manipulation, exploitation of artists via digital replicas.). It is noted that this issue is not unique to our methodology but prevalent in other human generative methodologies. Therefore, it is absolutely essential for current and future research in the field of 3D human generative modeling to address and reassess these considerations consistently.

