# OpenReview forum: "SyncHuman: Synchronizing 2D and 3D Generative Models for Single-view Human Reconstruction"
_NeurIPS.cc/2025/Conference — NeurIPS 2025 poster_

### Official Review · Reviewer_NEiA · 2025-06-16

**Clarity:** 3
**Significance:** 3
**Originality:** 3
**Rating:** 5
**Confidence:** 4

**Summary:**

This paper introduces method for reconstructing photorealistic 3D full-body humans. The key contributions are:

- Synchronized 2D multiview diffusion and 3D native diffusion, without SMPL priors, reconstruct high-quality clothed human mesh even under challenging human poses.
- Their unified decoder for 3DGS and mesh captures detail and fidelity from multiview features, and structural consistency from structured latents.

SyncHuman is evaluated on the CAPE and X-Humans dataset, demonstrating state-of-the-art performance in reconstructing clothed 3D human in both geometric and appearance quality.

**Questions:**

1. How long does SyncHuman take to reconstruct 3D human compared to baseline methods? (Weaknesses 1.)
2. The generative diffusion models can inpaint the missing regions under human-human or human-object interactions in MultiHuman or Hi4D dataset? (Weaknesses 2.)
3. The numerics of baseline methods in Tab. 1 are different from the ones reported in each paper. Additionally, SyncHuman is trained on larger training datasets: THuman 2.1, CustomHumans, THuman3.0, and 2K2K. The baseline methods are re-trained in same training setting or not?
4. SyncHuman is also trained without image condition to enable CFG during inference. How about the qualitative performance of unconditional generation?

**Ethical Concerns:**

["NO or VERY MINOR ethics concerns only"]

**Final Justification:**

The authors have addressed my concerns. I keep my original positive rating.

**Limitations:**

Computational costs might be one of the limitations.

**Paper Formatting Concerns:**

No.

**Quality:**

3

**Strengths And Weaknesses:**

### Strengths

1. The paper is well written and easy to understand.
2. They propose a novel framework that combines 2D multiview diffusion and 3D native diffusion for the first time.
3. SyncHuman achieves state-of-the-art performance in both geometric and appearance quality.
4. 2D-3D synchronization attention mechanism and MVGD are intuitive and effective for the performance, as shown in Tab. 2 and Tab. 3.

### Weaknesses

1. SyncHuman consists of three different diffusion: Multiview Diffusion, Structure Diffusion, and Structured Latents Diffusion. They do not report the computational costs, but the expensive computation might be one of the limitations.
2. The qualitative results demonstrate that SyncHuman is robust to self-occlusions. However, the input images are full-body images without inter-occlusions due to human-human or human-object interactions. It might be challenging to inpaint and generate missing regions without SMPL priors.

---

> ### Author Rebuttal · Authors · 2025-07-31
>
> We sincerely appreciate your efforts in reviewing our paper and providing insightful suggestions to help us improve our paper. We feel quite encouraged with your positive comments like "a novel framework", "state-of-the-art performance", and "effective designs". Hope the following responses address all your concerns.
>
> ### Q1: Inference time.
> On a single H800, the inference time is as follows: ours 38.57s vs Trellis  15.68s vs PSHuman 52.98s
> Our method is faster than PSHuman as it directly decodes a 3D shape without requiring additional differentiable rendering optimization. The slower speed compared to Trellis is due to our use of the 2D multi-view generation.
>
> ### Q2: Can the proposed method handle occlusions in human-human interaction (HHI) or human-object interaction (HOI)?
> We agree that extending our framework to HHI or HOI reconstruction is an interesting and important direction. However, our model is only trained on single-person data without HOI or HHI occlusions, so the model cannot handle such occlusions currently. This could be partially solved by introducing such occlusions during training. Meanwhile, extending to HHI or HOI reconstruction requires the model not only to guess unseen parts but also to ensure the guessed parts are compatible with the interacted subjects, which is out of the scope of our method. We think this could be an interesting future work, and thank you for pointing out such a promising future topic.
>
> ### Q3: Explain more about the experimental setting.
> For all baseline methods, we adopt their official model and codes for evaluation. We do not retrain all the baseline methods because retraining all baselines is too costly and unaffordable for us. Instead, we retrain two key baseline methods, PSHuman and Trellis, on the same training data as our method in experiments to demonstrate the effectiveness of our 2D-3D synchronization.
> The performances are different from their papers because the single-view reconstruction has a scale ambiguity. We adopt the following scale normalization strategy for evaluation, which may differ from previous methods. First, we align the centers of the reconstructed mesh and the ground truth mesh, then scale them so that the coordinate range of the longest axis is 1. All evaluations in our paper were obtained by re-running the inference using the open-source codes from the relevant papers, applying a unified evaluation method. We will open-source the related evaluation code to ensure the results are reproducible.
>
> ### Q4: Can our model do the unconditional generation?
> Yes. Following your suggestions, we have tested our model on the unconditional generation task. We find that our method produces different textured human meshes from different seeds with plausible textures and structures. However, we find that the generation quality is worse than the conditional generation from a single-view image. The reason may be that the conditional input images provide a precise denoising direction for the model, while the unconditional generation does not have a clear denoising direction, and thus the generations have worse structures and textures. We also tried this on the pretrained PSHuman model and the Trellis model. We observe similar quality degeneration from conditional generation to unconditional generation.

---

> ### Author Response · Authors · 2025-08-03
>
> We sincerely appreciate the time and effort you have dedicated to reviewing our manuscript. We have tried our best to provide responses to all of your questions and we hope our responses would address your concerns.
>
> As the discussion period nears its conclusion, we would be grateful for any final feedback you may have. Please don't hesitate to let us know if further clarifications would be helpful - we remain ready to provide additional details as needed.
>
> Thank you again for your valuable insights and constructive feedback throughout this process.

---

> > ### Comment · Reviewer_NEiA · 2025-08-04
> >
> > I appreciate the author response. They have addressed my concerns. I will keep my original rating.

---

> > > ### Author Response · Authors · 2025-08-05
> > >
> > > We are grateful for the time and effort you have devoted to reviewing our manuscript. Your expert comments and suggestions are valuable for improving our paper. In the revised version, we will make our best effort to address the points you raised.

---

### Official Review · Reviewer_Kbwv · 2025-06-29

**Clarity:** 3
**Significance:** 2
**Originality:** 3
**Rating:** 3
**Confidence:** 5

**Summary:**

This paper proposed a framework that combines 2D multiview diffusion and 3D native diffusion, enabling high-quality clothed human mesh reconstruction from single-view images.

Giving a single-view image, the method first generates 3D structures and multiview images. Then their use multi-view reconstructor to obtain 3D human textured meshes.

**Questions:**

Why did the author inject DINOv2 into MVGD? It requires an ablation study to demonstrate its effectiveness.

The ablation on 2D-3D synchronization attention may unfair. The author fine-tuned the compared methods on X-Humans dataset, which only contains 100 scans from less than 20 participants. However, the propose method is trained on Thuman, 2K2K and CustomHumans which contains about more than 2,500 scans. They should fine-tuned the compared methods on all training data.

The proposed method is incremental. The author prefers to combine PShuman, Trellis, and DINOv2. The 2D-3D cross spatial diffusion simply mixes the features of two branches together

**Ethical Concerns:**

["NO or VERY MINOR ethics concerns only"]

**Final Justification:**

Thanks to the author for providing a detailed answer to my question in the rebuttal. I have also read the opinions of other reviewers. I still think that the main contribution of this article is not solid enough, and there may be unfairness in the comparison of SOTA methods. Therefore, I will maintain my score.

**Limitations:**

yes

**Quality:**

2

**Strengths And Weaknesses:**

Strengths

The process of 2D-3D Cross-Space Diffusion and Multiview Guided Decoder improve the final results.

From the experimental results, it can be seen that the method proposed by the author outperforms the results of the comparative methods.


Weaknesses

Why did the author inject DINOv2 into MVGD? It requires an ablation study to demonstrate its effectiveness.

The ablation on 2D-3D synchronization attention may unfair. The author fine-tuned the compared methods on X-Humans dataset, which only contains 100 scans from less than 20 participants. However, the propose method is trained on Thuman, 2K2K and CustomHumans which contains about more than 2,500 scans. They should fine-tuned the compared methods on all training data.

The proposed method is incremental. The author prefers to combine PShuman, Trellis, and DINOv2. The 2D-3D cross spatial diffusion simply mixes the features of two branches together.

Repeat References. The [40&41], [71&72] references are same.

Missing references
DIFu: Depth-Guided Implicit Function for Clothed Human Reconstruction, CVPR 2023.

In L-269, a period is missing.

---

> ### Author Rebuttal · Authors · 2025-07-31
>
> We sincerely thank the reviewer for your valuable comments. We feel encouraged by your positive comments like "improved performance". Hope the following responses address all your concerns.
>
> ### Q1: What is the role of DINOv2 in MVGD?
> DINOv2 serves as a feature extractor for multiview images to inject multiview image features in the decoding process, which enhances the fidelity of details and structure of the 3D generation. We have conducted an ablation study by removing the multiview injection in Fig.8 and Tab.3. The "original decoder (tuned)"  without multiview image feature injection produces much more blurry results, while our injection module improves the fidelity and quality.
>
> ### Q2: Is the ablation of the 2D-3D synchronization attention unfair with the same training data?
> Yes. We fairly retrain all the baseline PSHuman and Trellis on the same training datasets in the ablation study and only conduct evaluation on the same unseen X-Humans dataset.
>
> ### Q3: The proposed method is incremental. The authors combine PShuman, Trellis, and DINOv2.
> To our knowledge, Synchuman is the first work that simultaneously generates both 2D and 3D representations for single-view 3D human reconstruction, achieving SOTA performance in terms of reconstruction precision and pose robustness. At its core, we integrate the strong prior of 3D native generative models and rich image-level details of 2D generative models, which benefit each other through effective 2D-3D attention. We further enhance the geometric details by injecting image features into the shape decoder. We believe this framework holds significant potential for advancing future 3D generation solutions, which enjoy the strengths of both well-engineered 2D generative models and emerging 3D native generative models.
>
> ### Q4: Repeat and missing references
> Thanks for pointing this out. DIFu improves occupancy prediction using depth priors, but it is limited in reconstructing plausible appearance. In contrast, our method leverages a pretrained 2d generative model to generate high-resolution multi-view color and normal images, achieving both detailed geometry and photorealistic appearance.  We will add this citation and discuss it in the related works.

---

> ### Author Response · Authors · 2025-08-03
>
> We sincerely appreciate the time and effort you have dedicated to reviewing our manuscript. We have tried our best to provide responses to all of your questions and we hope our responses would address your concerns.
>
> As the discussion period nears its conclusion, we would be grateful for any final feedback you may have. Please don't hesitate to let us know if further clarifications would be helpful - we remain ready to provide additional details as needed.
>
> Thank you again for your valuable insights and constructive feedback throughout this process.

---

> ### Author Response · Authors · 2025-08-05
>
> We are grateful for the time and effort you have devoted to reviewing our manuscript.With the discussion period coming to a close, we welcome any feedback. Let us know if you need further clarification—we’re here to clarify.

---

### Official Review · Reviewer_dgjc · 2025-06-29

**Clarity:** 2
**Significance:** 2
**Originality:** 2
**Rating:** 3
**Confidence:** 4

**Summary:**

This paper introduces SyncHuman, a method for reconstructing 3D textured human meshes from a single image by using 2D multiview diffusion and native 3D diffusion models. The framework uses a synchronization attention mechanism to bridge the two models, then uses a multiview-guided decoder to further inject fine details from generated 2D images into the 3D latent space. Experiments on CAPE and X-Humans show that SyncHuman achieves good performance and quality.

**Questions:**

See weaknesses.
The authors need to justify the numbers reported in the paper. Explain why it is worse than the reported numbers by a significant margin (larger than the performance improvements). Is it an implementation error by the authors or evaluation bugs? In general, I have completely lost trust in the numbers I see in the paper, and the qualitative results of the previous method could be downplayed as well (compared to the qualitative results reported in their own paper).

**Ethical Concerns:**

["NO or VERY MINOR ethics concerns only"]

**Final Justification:**

The authors' rebuttal has addressed most of my questions. I raise my final rating to borderline reject. The concerns remain about the marginal improvements from the proposed components. The provided visual ablation figures lack clear failure modes or patterns corresponding to the main novelty of the paper. The paper lacks experimental analysis about why these new designs lead to consistent improvements and what practical problems have been "fundamentally" solved.

Besides, I am not satisfied with the authors' original reckless attitude toward the ethical checklist. Acting ignorantly when working in a highly sensitive research domain is unacceptable. I am not very happy about this situation. Such a course of action can be waived with a simple apology. The acceptance of this paper introduces another group of reckless people getting into the community, encouraging more similar misbehavior.

However, considering my role is not ethics reviewer, I will not fight for rejection on this.

**Limitations:**

As disclosed in the paper, it is known that multi-view diffusion models are trained on idealized shadowless objects, so they cannot handle strong lighting. This limits its real-world applications. As mentioned in the weaknesses, the method may suffer from overly smoothened textures, but the authors did not mention relevant weaknesses.

**Quality:**

2

**Strengths And Weaknesses:**

Strengths:
1. The qualitative results are somewhat intriguing.
2. The method avoids using SMPL, which has several known issues listed in the paper.

Weaknesses:
1. While the synchronized attention implementation is somewhat novel, the idea of enhancing geometry using a multi-view diffusion model is a widely adopted idea. The approach is primarily a well-engineered system combining two pretrained diffusion models.
2. The numbers reported in Table 1 are different from the original numbers reported in their own paper (e.g., PSHuman). These significant downplaying remarks the potential and intentional unfavorable calibration to the evaluation, questioning the trustworthiness of certain baselines that were not evaluated on the benchmark (e.g., TRELLIS). The only interesting part of the paper is its outstanding performance shown in Table 1, but the numbers are not really reported in a responsible way. The Chamfer distances reported in the results are all unreasonably bad, which questions the generalization of the SMPL failures cases reported by the authors.
3. The failure cases of SIFU and PSHuman in Figure 6 are significantly worse than it supposed to be. Certain distortions and limb discontinuities are so severe that iit s unbelievable.
4. The ablation study lacks depth, and the improvements are incremental.
5. 2D diffusion models are not very good at generating thin geometry (such as limbs and fingers), or maintain the semantic correctness (such as generating three legs, especially common in legacy StableDiffusion models). Do the authors observe similar problems? How is it addressed?
6. The cross-view diffusion models are not robust in maintaining the geometric consistency of fine textures, such as hairs and wrinkles. Do the authors observe overly smoothened textures?
7. The authors are extremely reckless in completing the checklist. I don't think the authors qualify to respond yes to "7. Experiment statistical significance." The response to "8. Experiments compute resources" clearly does not read the guidelines. The answer to 10, 11, 15, simply disregards the human subject data used during the model training. Some of the training datasets do not have subject's consent. I typically don't pay too much attention to the checklist as people should have known what they are doing, but it is apparently not the case in the submission.

---

> ### Author Rebuttal · Authors · 2025-07-31
>
> Thank you for your efforts in reviewing our paper and providing insightful comments. We feel encouraged by your positive comments like "intriguing qualitative results". Hope the following responses address all your concerns.
>
> ### Q1: The novelty is limited because some papers already apply multiview images to improve 3D generation.
> Our novelty lies in the effective synchronized 2D-3D generation, which shows great improvements in single-view 3D human reconstruction. While some works employ multiview generation to enhance 3D geometry, they typically adopt a sequential scheme: generating multi-view 2D images to feed into a 3D generative model, or generating a 3D model to guide the multiview generation for refinement. Such sequential methods inherently suffer from error accumulation; inaccuracies in the initial stage, like inconsistent multiview images, often limit the effectiveness of subsequent 3D generation or refinement. In contrast, our approach generates multiview images and 3D models simultaneously in a synchronized manner, enabling mutual reinforcement. Therefore, SyncHuman improves both the geometric fidelity of 3D outputs and the cross-view consistency of generated images.
>
> ### Q2: The reported performances are different from those in baseline papers (e.g. PSHuman).
> The performances are different from their papers because the single-view reconstruction has a scale ambiguity. We adopt the following scale normalization strategy for evaluation, which may differ from previous methods. We align the centers of the reconstructed mesh and the ground truth mesh and then scale them so that the coordinate range of the longest axis is 1. All evaluations in our paper were obtained by re-running the inference using the open-source codes from the relevant papers, applying a unified evaluation method. We will open-source the related evaluation code to ensure the results are reproducible.
>
>
> ### Q3: The results for PSHuman and SIFU are unreasonably bad.
> We compare against PSHuman and SIFU fairly with the same evaluation setting.
> Our reported metric numbers for PSHuman are better than the original paper. Note that in Table 1 of the PSHuman paper, the setting "w/ SMPL-X prior" means using ground-truth SMPL-X parameters, while "w/o SMPL-X prior" means using estimated SMPL-X parameters. Our method does not rely on GT SMPL, so we are comparing against the upper part ("w/o SMPL-X prior") of Table 1 in the PSHuman paper.
> PSHuman and SIFU heavily rely on SMPL prediction, and the SMPL errors will lead to severe reconstruction artifacts, especially for humans with complex poses in Fig.6. In contrast, our 3D generative models generate accurate 3D structures on such poses. Note that Human3Diff in Fig.6 does not utilize inaccurate SMPL estimation either and thus also shows a better structure.
>
> ### Q4: Insufficient ablation, improvements are incremental.
> We have conducted ablations to validate the effectiveness of our designs in Fig.7, Fig.8, Tab.2, and Tab.3. The results show that our 2D-3D attention effectively synchronizes 2D and 3D generation with a better quality, while our MVGD significantly enhances the geometry detail and fidelity.
>
> ### Q5: Can the proposed method reconstruct thin geometry like fingers?
> Yes. As shown in Rows 1-3 of Fig. 1 in the supplementary material, our method succeeds in reconstructing fingers. Though fingers are small, they can still be reconstructed from multiview generation. We agree that more accurately reconstructing fingers is still an open problem and an interesting future topic, which can be combined with parameteric hand shapes like MANO.
>
> ### Q6: Will the multiview generation produce semantically incorrect results like 3 legs?
> The vanilla multiview generation may occasionally produce such artifacts. However, by integrating 3D generation, which effectively learns human structural priors, our method significantly reduces these errors and maintains semantic correctness in the generated outputs.
>
> ### Q7: Can the proposed method reconstruct fine detailed textures such as hairs and wrinkles?
> Yes. As shown in Rows 1-3 of Fig.5 in the main paper, our method succeeds in reconstructing the wrinkles on the clothes. For the hair textures, as shown in Rows 2 and 3 of Fig.6 in the main paper, our method can get reasonable results for the detailed hair textures. We agree that accurately reconstructing all the textures and wrinkles still remains a challenging task, especially within our ill-posed single-view reconstruction setting, and our method still shows some blurry textures in reconstructing the exact strands of hair. Our method already shows significant improvements over previous methods by a large margin in terms of details and fidelity. We will add the oversmooth problem of hair textures in the revision.
>
> ### Q8: The checklist is treated recklessly.
> Thank you for pointing out that some checkpoints about ethical issues may be answered improperly. We have carefully checked the list, but we had a misunderstanding that if we all use existing public human datasets, then there would not be an ethical issue. We will revise this part according to your suggestion and discuss more about the ethical issues in the revision.
>
> ### Q9: The method cannot handle strong lighting, which limits its real-world applications.
> In the supplementary video, we provide high-quality results on real-world in-the-wild single-view images, and both our evaluation datasets, CAPE and X-Humans, are real-world datasets, which all demonstrate the potential of our method in real-world applications.
> We have discussed the limitations that our method could produce some white artifacts in textures when strong highlights exist, but still have a reasonable geometry reconstruction. Handling such highlights may require rendering relightable 3D avatars under different lighting conditions, but such data is still scarce, and almost none of the existing single-view avatar reconstruction methods could effectively handle this problem. This could be a promising future research topic.

---

> ### Author Response · Authors · 2025-08-03
>
> We sincerely appreciate the time and effort you have dedicated to reviewing our manuscript. We have tried our best to provide responses to all of your questions and we hope our responses would address your concerns.
>
> As the discussion period nears its conclusion, we would be grateful for any final feedback you may have. Please don't hesitate to let us know if further clarifications would be helpful - we remain ready to provide additional details as needed.
>
> Thank you again for your valuable insights and constructive feedback throughout this process.

---

> > ### Comment · Reviewer_dgjc · 2025-08-04
> >
> > Sincerely, thanks for the authors' clarifications. The rebuttal has addressed most of my questions, and I will raise my final rating to weak rejection.
> > The concerns remain about the marginal improvements from the proposed components. The provided visual ablation figures lack clear failure modes or patterns corresponding to the main novelty of the paper. The paper lacks experimental analysis about why these new designs lead to consistent improvements and what practical problems have been "fundamentally" solved.

---

> ### Author Response · Authors · 2025-08-05
>
> ####
> Thank you so much for your constructive comments and for helping us improve our paper. For your further questions, we hope the following responses and clarifications could address your concerns.
>
> **Q1: What practical problems have been "fundamentally" solved?**
>
> Our core contribution lies in a novel framework of 2D-3D joint generation, which fundamentally improves the fidelity, details, and structure plausibility of single-view human reconstruction.
>
> 1) In comparison with 3D native diffusion models like Trellis and Hunyuan, our method greatly improves the fidelity and reconstruction details, as evidenced by Fig.7 (comparison with Trellis) and Fig. 1 of the supplementary material (comparison with Hunyuan).
>
> 2) In comparison with multiview generative models like PSHuman, our method improves the structural plausibility, as evidenced by Fig. 5 and Fig. 6.
>
> **Q2: Why do these new designs lead to consistent improvements?**
>
> 1) 3D native generative models lack pixel-wise alignment between the 3D generation results and the input images, while our method aligns both the input and generated images with 3D generation results, and further benefits from the fine-grained details typically present in high-resolution image generation.
>
> 2) 2D multiview generative models, like PSHuman, often rely on an estimated SMPL mesh to improve multiview consistency and generate a human mesh, which often generate structurally implausible 3D human meshes due to the errors of SMPL estimation. In comparison, our method effectively utilizes the 3D native generation method to provide a coarse structure for multiview 2D generation, which improves not only the structural plausibility but also the multiview consistency of the generated images.
>
> **Q3: How does the ablation study demonstrate the effectiveness of the main novelty?**
>
> We achieve our 2D-3D joint generation framework by proposing 2D-3D synchronization attention and a multi-view guided decoder (MVGD). The quantitative comparisons (Tab.1) among baselines have shown the superiority of our novel framework with a Chamfer Distance of 0.9127 (ours) vs. 1.3726 (PSHuman) vs. 1.6573 (SIFu) vs. 2.0877 (Trellis), which owes to our structural plausibility. We also conduct comprehensive ablation studies in Fig.7, Fig.8, Tab.2, Tab.3, and our response to Reviewer Pinc (Q3). Especially from the visual comparison, we can see that our method clearly has a better fidelity and details than 3D native generative model only (Trellis), and also has a much better structural plausibility than multiview generative model (PSHuman) with incorrect SMPL estimations.
>
> We will further clarify the above discussion in the revision following your suggestions and comments. Thank you!

---

### Official Review · Reviewer_6GgC · 2025-06-30

**Clarity:** 3
**Significance:** 2
**Originality:** 2
**Rating:** 4
**Confidence:** 4

**Summary:**

This paper presents a method to reconstruct detailed 3D human from single RGB image. The idea is to combine the texture details from multi-view diffusion model with structure priors from TRELLIS. They introduce cross attention module to synchronize the 2D features from multi-view diffusion models and 3D features from the sparse grid of TRELLIS. The synchronization is done via pixel aligned orthographic projection. They further inject the multi-view features into sparse voxels before final decoding to enhance the details. Experiment resutls on two datasets demonstrate the effectiveness of the proposed pipeline. Ablations also show the importance of the proposed modules.

**Questions:**

- How would this model compare to Hunyuan3D 2.0? As a more advanced version of TRELLIS, HY3D shows impressive details and on par texture details. It would be interesting to see if an off-the-shelf general object reconstruction model can already work well for the specific human reconstruction task. If not, what is the missing gap?

- Strangely large errors (P2S, CD) from TRELLIS while TRELLISS (fine tuned) is much better, table 1-3. Did you do proper alignment (e.g. SiTH [1]) between reconstruction and GT mesh before reporting the numbers? Or please explain the gap between trellis and trellis-fine tuned.

[1]. SiTH alignment for evaluation: https://github.com/SiTH-Diffusion/SiTH/blob/main/EVALUATE.md#1-single-view-human-3d-reconstruction-benchmark

**Ethical Concerns:**

["NO or VERY MINOR ethics concerns only"]

**Final Justification:**

The author response has addressed my concerns. I will keep my original rating as weak accept.

**Limitations:**

yes

**Quality:**

3

**Strengths And Weaknesses:**

**Strength**
- The overall writing flow is clear.
- The proposed method works well on the evaluated benchmark datasets.
- Ablation studies prove the effectiveness of the proposed modules.

**Weakness**
- Originality: the method overall is not very novel. The synchronized attention is pretty much the same as SyncDreamer, even the name is the same pattern -- SyncHuman. Even the MVGD module is also similar to SyncDreamer where they inject 3D volume feature to 2D multi-view but here it is in the opposite direction: 2D to 3D.
- Clarity: Some method details are not clear. Is the 2D-3D synchronized attention done at each reverse sampling step? Eq. 3: flow matching loss is used here but in all previous text the term 'diffusion models' are used. Although mathematically they are very similar, but it is important to clarify them to be precise.
- Clarification of the experiment setup: L240-'subset of X-humans', is it the same test set used in table 1? Also in table 2 and table 3, image metrics like PSNR/SSIM are reported, are they the generated 2D multi-view images or re-rendering of the reconstructed 3D? Although the end goal is to have better 3D, but it would be interesting to see if this cross attention mechanism would also improve the 2D multi-view diffusion model. Would this also improve the predicted multi-view normal?

---

> ### Author Rebuttal · Authors · 2025-07-31
>
> We are grateful for your valuable feedback. We feel encouraged by your positive comments like "works well on the evaluated benchmark datasets". We hope the following responses address all your concerns..
>
> ### Q1: Differences from SyncDreamer
> Our method fundamentally differs from SyncDreamer in the following two aspects.
> First, the synchronization subjects are totally different. SyncDreamer synchronizes the generation of multiview 2D images, whereas our method synchronizes the 2D generative model and the 3D native generative model. Our method demonstrates that simultaneously generating multiview images and 3D representations benefits each other and greatly improves the 3D generation quality. We inject information in both directions, 2D to 3D and 3D to 2D.
> Second, the functionality and design of the volume in our method are fundamentally different from those in SyncDreamer. SyncDreamer constructs a feature volume to share information among different views. In contrast, in our method, the volume is a meaningful 3D representation generated from noise.
> We have discussed SyncDreamer and related multiview diffusion methods in the paper, and will further clarify this according to your suggestion.
>
> ### Q2: Is the 2D-3D synchronization attention done at each reverse sampling step?
> Yes. The 2D-3D synchronization attention is used at all denosing steps to maintain consistency between 2D and 3D and improve the quality of both.
>
> ### Q3: Be more careful about the terms flow matching and diffusion.
> Our model adopts flow matching to train both 2D and 3D generative models. Thanks for pointing this out, and we will rectify this in the revision.
>
> ### Q4: L240-'subset of X-humans', is it the same test set used in Table 1?
> Yes, the ablation study uses the same X-Humans subset as Table 1.
>
> ### Q5: How do you evaluate image quality?
> The metrics for image quality in Tables 2 and 3 are computed from the renderings of 3D meshes for all methods. For each mesh, we render four views with a 768 resolution and report the average PSNR, SSIM, and LPIPS.
>
> ### Q6: Would the 2D-3D attention improve multiview generation quality?
> Yes. Following your suggestion, we additionally report the quality of intermediate multi-view generation.
> | Method          | PSNR↑   | SSIM↑  | LPIPS↓ |
> |-----------------|--------|--------|--------|
> |w/o att (color)  | 23.328 | 0.877  | 0.078  |
> |ours (color)    | **24.027** | **0.894** | **0.070** |
> |w/o att (normal) | 22.851 | 0.866  | 0.097  |
> | ours (normal)   | **23.439** | **0.882** | **0.087** |
> Through our 2D-3D attention, the 3D representations effectively associate multiview images for better quality and multiview consistency. We will add this in the revision.
>
> ### Q7: Comparison with Hunyuan3D
> In Fig.1 of the supplementary material, we conduct a qualitative comparison with the commercial model Hunyuan3D-2.5 (not open-sourced yet). Although Hunyuan3D-2.5 could produce plausible structures, it struggles with details and appearance fidelity, which is a common issue for existing native 3D generative models.
> This gap arises because Hunyuan3D only applies cross-attention layers to associate the input image and 3D shape, lacking pixel-aligned information. In contrast, our 2D-3D synchronized attention and MVGD incorporate pixel-aligned features, significantly improving structural and visual fidelity.
>
> ### Q8: Why is there a performance gap between Trellis and tuned Trellis? Are the generated meshes aligned with GT?
> For both tuned and pretrained Trellis models, we apply mesh normalization and ICP registration to align the output meshes with the gt meshes to ensure a fair comparison. The performance gap between the original and tuned Trellis stems from two factors. First, while the original Trellis is trained on diverse datasets containing only limited human meshes, our tuned version is specifically optimized using high-quality 3D human scans. Second, the original model generates meshes in a canonical space without aligning with the input images. In contrast, our fine-tuned Trellis model generates meshes in a coordinate system aligned with the input views. This alignment correlates the 3D shape with the input image strongly and thus significantly improves both accuracy and spatial consistency. We will add this discussion to the revision following your suggestion.

---

> > ### Comment · Reviewer_6GgC · 2025-08-04
> > **thank you for the response**
> >
> > I appreciate the author response. They have addressed my concerns. I will keep my original rating.

---

> > > ### Author Response · Authors · 2025-08-05
> > >
> > > We are grateful for the time and effort you have devoted to reviewing our manuscript. Your expert comments and suggestions are valuable for improving our paper. In the revised version, we will make our best effort to address the points you raised.

---

> ### Author Response · Authors · 2025-08-03
>
> We sincerely appreciate the time and effort you have dedicated to reviewing our manuscript. We have tried our best to provide responses to all of your questions and we hope our responses would address your concerns.
>
> As the discussion period nears its conclusion, we would be grateful for any final feedback you may have. Please don't hesitate to let us know if further clarifications would be helpful - we remain ready to provide additional details as needed.
>
> Thank you again for your valuable insights and constructive feedback throughout this process.

---

### Official Review · Reviewer_Pinc · 2025-07-02

**Clarity:** 2
**Significance:** 3
**Originality:** 3
**Rating:** 4
**Confidence:** 5

**Summary:**

This paper proposes SyncHuman, a novel approach for single-view 3D clothed human generation. The method effectively combines the strengths of 2D multiview diffusion models and native 3D diffusion models through pixel-aligned 2D-3D synchronization attention and a multiview feature injection strategy. After fine-tuning on a collected dataset, the proposed method achieves impressive photorealistic 3D full-body human reconstruction from a single image.

**Questions:**

Specifically, I have a few questions regarding the paper and would appreciate it if the authors could provide clarification:
1. During the 2D to 3D attention process, how are the resulting features added into the original 3D voxel features? Are the features averaged, concatenated, or combined in another way? Is this cross-attention mechanism applied at every diffusion timestep, or only at selected timesteps? Similarly, how are features incorporated during the 3D to 2D attention process? More detailed explanations of these mechanisms would be helpful.

2. During joint training of the 2D and 3D models, are all models trained together, or are some model weights frozen? Is MVGD involved in the joint training process? Please provide more details on the training procedure, such as the specific process of the training, the amount of data used, the number of training epochs, and relevant hyperparameters.

3. In the ablation experiments for the effectiveness of 2D-3D cross-space attention, could the authors compare the generation results of the 2D diffusion model before and after applying cross-attention? Theoretically, the 2D diffusion model with cross-attention should produce more cross-view consistent multi-view results compared to the original model. Similarly, does the 3D diffusion model show corresponding improvements after cross-attention is applied? Additional experiments to support these claims would strengthen the paper.

4. It appears that two decoder branches were trained: 3DGS and mesh. Which branch’s results are shown in the experimental section? How significant are the differences between the two branches? It would be helpful to present results from both branches for comparison.

5. In Fig. 6, although the overall pose and texture of the generated results are consistent with the input image, there are still noticeable inconsistencies in many details, such as the face shape in the first and third examples, and the skin tone in the last example. Additionally, from the video results, the generated humans still suffer from blurriness and white artifacts.

6. The paper is missing several relevant references, including:
1)	ConTex-Human: Free-View Rendering of Human from a Single Image with Texture-Consistent Synthesis
2)	HumanRef: Single Image to 3D Human Generation via Reference-Guided Diffusion
3)	DINAR: Diffusion Inpainting of Neural Textures for One-Shot Human Avatars
4)	ZeroAvatar: Zero-shot 3D Avatar Generation from a Single Image

**Ethical Concerns:**

["Major Concern: Data privacy, copyright, and consent"]

**Final Justification:**

The authors’ response has largely addressed my concerns. I agree to accept the paper on the condition that the supplementary experimental results, especially the visualizations for Q3, are included in the camera-ready version.

**Limitations:**

No, the paper does not provide an explanation for the artifacts present in the results. Perhaps a related discussion could be added to the Limitations section.

**Quality:**

3

**Strengths And Weaknesses:**

Strengths:
The approach successfully integrates the advantages of both 2D and 3D diffusion models. The experimental results convincingly demonstrate the effectiveness of the proposed design.

Weaknesses:
The paper is missing relevant citations and lacks technical details and necessary discussion and analysis.

---

> ### Author Rebuttal · Authors · 2025-07-31
>
> We sincerely thank the reviewer for these insightful comments and positive words like "successful combination of 2D-3D diffusion models" and "convincingly demonstrate the effectiveness". Hope the following responses address all your concerns.
>
> ### Q1: Details of 2D-3D Cross-Space Diffusion
> The synchronization attention is used at all denosing steps to maintain the consistency and improve the quality of the two branches. Both the 2D-to-3D and 3D-to-2D attention are implemented via cross-attention (Fig.4). In 2D-to-3D attention,  each voxel serves as a query, and the concatenation of corresponding orthogonal sample embeddings on multiview images is used as keys and values. In 3D-to-2D attention, each pixel-level embedding serves as a query, and the corresponding 3D voxel embedding in projected 3D space is the key and value. The resulting features are processed with a zero-initialized MLP and are added back to the denoising branch. The detailed implementation is elaborated on lines 35-62 of the supplementary material, and we will add more to make it clearer in the revision.
>
> ### Q2: Add more training details
> The 2D multiview generative model is initialized with SD2.1, the 3D structure generative model, and MVGD are initialized from Trellis. The training process includes two stages. At the first stage, the 2D multiview generative model ,the 3D structure generative model and 2D-3D synchronization attention are jointly tuned by tuning all parameters. At the second stage, we train MVGD. The training dataset is composed of publicly used 3d human scans, including THuman2.1, CustomHumans, THuman3.0, and 2K2K. We have included implementation details in lines 1-34 of the supplementary materials and will add more in the revision. We will release all code for the reproducibility of our method.
>
> ### Q3: Could the 2D-3D synchronization attention improve the multiview generation quality or the 3D structure quality?
> In our ablation study, we have reported the rendering quality and the geometry accuracy of the final reconstructed 3D meshes with or without 2D-3D synchronization attention. Following your suggestions, we additionally report the quality of intermediate multi-view generation and 3D structure generation.
> IOU of the 3D structure generation:  0.5907 (with 2D-3D synchronization attention) vs. 0.4813(without 2D-3D synchronization attention).
> Color and normal image quality improvements by 3D-2D attention:
> | Method          | PSNR↑   | SSIM↑  | LPIPS↓ |
> |-----------------|--------|--------|--------|
> |w/o att (color)  | 23.328 | 0.877  | 0.078  |
> |ours (color)    | **24.027** |**0.894** | **0.070** |
> |w/o att (normal) | 22.851 | 0.866  | 0.097  |
> | ours (normal)   | **23.439** | **0.882** |**0.087** |
> Because generating 3D structures or multiview images from single-view inputs has ambiguity, the generation results are not exactly the same as the ground-truth. However, our 2D-3D attention could produce results more similar to GT. This demonstrates that our 2D-3D synchronization attention could benefit both branches to improve the multiview generation quality and 3D structure quality. We will add this result to the revision following your suggestions.
>
> ### Q4: How to combine the 3DGS and mesh decoding results
> We follow Trellis to bake the GS renderings onto the generated mesh because 3DGS has better rendering quality, while meshes have better surface geometry. We have included this in L198-199 and will explain it more to make it clearer.
>
> ### Q5: Face inconsistency in Fig. 6 and highlight artifacts in videos.
> In Fig.6, we have rotated the 3D models a little bit to show they are in 3D space, which leads to inconsistency with the input image due to the viewpoint changes of the faces. We will clarify this in the revision.
> We agree that our method is currently not robust to highlights, and we have included this in our limitations discussion. When a white highlight appears at the edges of the human in the input image, our method may generate some white patterns in the back view. One possible solution may be to add some augmentations to improve the robustness, which requires scarce relightable 3D avatars to render under different illumination conditions.
>
> ### Q6: Missing references.
> Thanks for pointing this out. ConTex-Human, HumanRef, and ZeroAvatar are SDS-based methods that rely on time-consuming per-instance optimization. DINAR heavily depends on SMPL estimation and UV-based texture generation, which restricts its ability to handle loose clothing. In contrast, our approach reconstructs 3D humans using a generative model in a SMPL-free manner, significantly improving both efficiency and robustness. We will add these citations and discuss them in the related works.

---

> > ### Comment · Reviewer_Pinc · 2025-08-04
> > **Thanks for the authors' response**
> >
> > The authors’ response has largely addressed my concerns. I agree to accept the paper on the condition that the supplementary experimental results, especially the visualizations for Q3, are included in the camera-ready version. For now, I will maintain my previous rating.

---

> > > ### Author Response · Authors · 2025-08-05
> > >
> > > We are grateful for the time and effort you have devoted to reviewing our manuscript. Your expert comments and suggestions are valuable for improving our paper. In the revised version, we will make our best effort to address the points you raised, and we will include the visualizations for Q3.

---

> ### Author Response · Authors · 2025-08-03
>
> We sincerely appreciate the time and effort you have dedicated to reviewing our manuscript. We have tried our best to provide responses to all of your questions and we hope our responses would address your concerns.
>
> As the discussion period nears its conclusion, we would be grateful for any final feedback you may have. Please don't hesitate to let us know if further clarifications would be helpful - we remain ready to provide additional details as needed.
>
> Thank you again for your valuable insights and constructive feedback throughout this process.

---

### Decision · Program_Chairs · 2025-09-17

**Decision:**

Accept (poster)

**Comment:**

The paper tackles the problem of 3D human generation from a single image. SyncHuman relies on a combination of multi-view and 3D diffusion; it introduces a 2D-3D synchronized attention mechanism to make the two branches informed of each other, and a multiview feature injection, which propagates multi-view features in the 3D voxels, to enhance the details.

At the beginning of the discussion phase, the opinions on the paper were mixed. On the bright side, there have been experimental results and comparisons with the state of the art, but there were also concerns about the methodological contribution, doubts in some comparisons with the values reported in the original paper, and unclear aspects in the experimental setting. The discussion has been extensive, but positions remained mostly unchanged, with a little upgrade of a rejection to borderline (dgjc). A reviewer raised concerns on the ethical side, and two ethical reviewers engaged, assessing that such weaknesses can be fairly easily addressed in the final version of the manuscript.

After the discussion, the score situation remained uncertain, with three reviewers (NEiA, Pinc, and 6GgC) leaning toward acceptance, while two reviewers (dgjc and Kbwv) argued for rejection. The remaining concerns are:

1) Significance of the contribution and novelty (dgjc, Kbwv): Reviewers believe the work follows a road already paved by previous works, which combine multi-view generations and 3D. The Authors see in the idea of synchronizing the 2D and 3D generation the key novelty of their approach. While previous works in general generate the two in sequences (e.g., obtaining the 3D as reconstruction from the multi-views, or updating the multi-views using a 3D reconstruction), they propose to learn them jointly.

2) Motivation behind improvements of the contribution (dgjc): The Reviewer believes that further motivation behind the improvements should be provided. The Authors argue that recent methods lack a clear alignment between the images and the 3D, while the proposed approach proposes to align the 2D multi-views to the volume and orthogonally project the feature to obtain a 3D representation that is actually aligned with the views.

3) fairness w.r.t. SoTA comparison (dgjc, Kbwv): The Reviewers have doubts about fairness in baseline training, and comparison with other methods, especially due to reported qualitative results. Authors argue that they train all the baselines on the same data, and results achieved by replicating SIFU and PSHuman do not use ground-truth SMPL, and numbers are hence in line with the expectation.

The AC recognizes that the paper does not introduce any dramatic paradigm change. However, the proposed approach explores a simple yet effective idea that could be of inspiration beyond the human case (e.g., 3D/4D generation of generic objects). The AC finds the reason behind error accumulation in sequential steps reasonable. Also, implementing such an idea required introducing components to connect 2D and 3D generation, which could be useful for future work. Finally, the paper shows that pose estimation of SMPL might be inaccurate; although further details would be useful to fully assess the method quality (e.g., reporting also the case of baselines using GT SMPL, as an "unfair" upper-bound of their performance; reporting clearly if all the baselines rely on the same pose estimators and which they are), the provided evidence is convincing in demonstrating SyncHuman superiority.

Due to these reasons, the AC leans toward acceptance, while encouraging the authors to:
1) Better state the motivation and the intuition behind the work
2) Clarify all the experimental and ablation details that have been pointed out by the reviewers
3) Take into serious consideration the ethical concerns, and integrate a sincere discussion in the paper.